# Disentangled Hierarchical VAE for 3D Human-Human Interaction Generation

**Zichen Geng**[1]   **Zeeshan Hayder**[2]   **Bo Miao**[3]   **Jian Liu**[4]   **Wei Liu**[1]   **Ajmal Mian**[1]

[1]Department of CSSE, The University of Western Australia, Crawley WA 6009, Australia
[2]Data61, CSIRO, Acton ACT 2601, Australia
[3]Australian Institute for Machine Learning, Adelaide University, Adelaide SA 5000, Australia
[4]NERC-RVC, Hunan University, Changsha 410012, China

`zen.geng@research.uwa.edu.au`,  `zeeshan.hayder@data61.csiro.au`
`bo.miao@adelaide.edu.au`,  `jianliu@hnu.edu.cn`
`{wei.liu, ajmal.mian}@uwa.edu.au`

## Abstract

Generating realistic 3D Human-Human Interaction (HHI) requires coherent modeling of the physical plausibility of the agents and their interaction semantics. Existing methods compress all motion information into a single latent representation, limiting their ability to capture fine-grained actions and inter-agent interactions. This often leads to semantic misalignment and physically implausible artifacts, such as penetration or missed contact. We propose Disentangled Hierarchical Variational Autoencoder (DHVAE) based latent diffusion for structured and controllable HHI generation. DHVAE explicitly disentangles the global interaction context and individual motion patterns into a decoupled latent structure by employing a CoTransformer module. To mitigate implausible and physically inconsistent contacts in HHI, we incorporate contrastive learning constraints with our DHVAE to promote a more discriminative and physically plausible latent interaction space. For high-fidelity interaction synthesis, DHVAE employs a DDIM-based diffusion denoising process in the hierarchical latent space, enhanced by a skip-connected AdaLN-Transformer denoiser. Extensive evaluations show that DHVAE achieves superior motion fidelity, text alignment, and physical plausibility with greater computational efficiency. The official implementation is publicly available at: https://github.com/ZenGengChin/dhvae-official

## 1 Introduction

Humans naturally coordinate with each other through timed and spatially aligned actions, such as shaking hands, dancing, playing sports, or passing objects. Representing and generating such human-human interactions (HHIs) in 3D is a core challenge in embodied AI, with broad impact on virtual character animation, human-robot collaboration, and embodied communication. Given a simple natural language prompt like "Person A hands an object to Person B," generating motion sequences for both agents that are semantically aligned, temporally coherent, and physically plausible remains an open challenge.

Early progress in motion generation largely focused on single-agent synthesis, where approaches like T2M-GPT Zhang et al. (2023), MotionGPT Jiang et al. (2024a), and MDM Tevet et al. (2023) successfully model long-range temporal dynamics conditioned on text. However, extending these techniques to multi-agent interactions is non-trivial. HHI generation presents unique challenges: it requires modeling synchronized dynamics between multiple agents, capturing both mutual awareness and individual autonomy, and handling diverse interaction semantics ranging from high-level coordination to fine-grained local motions. Recent advances in generative modeling, particularly latent diffusion models (LDMs) Rombach et al. (2021), have shown impressive performance in synthesizing high-dimensional data across domains, including human motion. By operating in a compressed latent space, these models enable efficient learning and scalable generation. MLD Chen et al. (2023) is the first to adopt this paradigm for single-agent motion generation, achieving good results through temporal-aware latent diffusion.

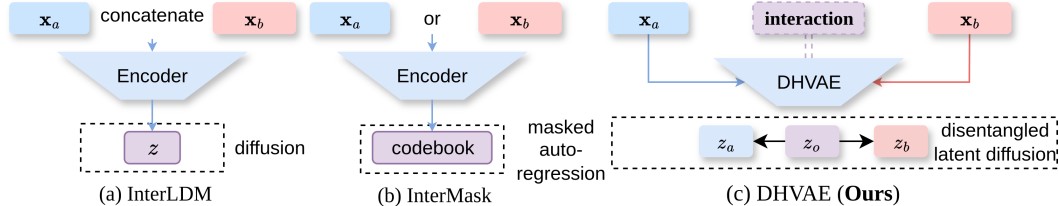

Figure 1: (a) InterLDM Li et al. (2025), (b) InterMask Javed et al. (2025) encode all motion information into a single latent. (c) Our encodes individual motions and interactions into separate disentangled latents.

Extending LDMs to human-human interaction, however, remains underexplored. A few recent attempts, such as InterLDM Li et al. (2025), have begun applying this idea to HHI. As illustrated in Fig. 1 (a), these approaches encode both agents into a flat, unified latent representation and apply diffusion in this joint space. While such a design enables synchronized modeling, it entangles agent identity with interaction context, leading to limited expressivity and degraded realism, particularly in cases requiring fine-grained coordination or distinct agent behaviors.

Moreover, as illustrated in Fig. 1 (b), models such as InterMask Javed et al. (2025) encode motion within a unified latent space shared across agents. While this approach has achieved state-of-the-art (SOTA) performance, it lacks explicit modeling of global interactions between agents, often resulting in physically implausible outcomes, e.g., hand penetration or missed contact in tasks like "two people shake hands." These failure cases underscore two fundamental limitations in current methods: the absence of structured interaction representations and limited control over contact realism.

We propose a new paradigm for text-conditioned human-human interaction generation: Disentangled Hierarchical Variational Autoencoder (DHVAE) paired with structured latent diffusion. Our key innovation is to explicitly disentangle the HHI representation into three levels: (1) $\mathbf{z}_a$, modeling Person A's individual motion; (2) $\mathbf{z}_b$, modeling Person B's individual motion; and (3) $\mathbf{z}_o$, a shared latent capturing the global interaction context. To fully leverage this hierarchy, we introduce a Co-Transformer module that jointly encodes mutual awareness and preserves individual agent identity. Beyond architectural design, we further enhance the learning of the interaction representation $\mathbf{z}_o$ using a contrastive learning strategy. We introduce a simple yet effective approach to construct positive and negative HHI pairs, imposing prior-based supervision on $\mathbf{z}_o$ to encourage it to encode meaningful and physically plausible interactions. This design directly addresses the lack of prior-based modeling for physically realistic contact interactions, a key limitation in prior works. The structured latents $\{\mathbf{z}_o, \mathbf{z}_a, \mathbf{z}_b\}$ are passed through an AdaLN Peebles & Xie (2023) Transformer-based denoiser trained using a Denoising Diffusion Implicit Model (DDIM) Song et al. (2021) process in the latent space. To address the scale imbalance and structural heterogeneity between $\mathbf{z}_o$, $\mathbf{z}_a$, and $\mathbf{z}_b$, we introduce segment positional encoding (SPE) to reflect each token's role in the interaction, and token scaling to calibrate feature magnitudes across latent groups. We also adopt skip connections for the AdaLN Transformer to stabilize training while allowing flexible conditioning.

Overall, we propose a disentangled and controllable framework for generating HHI motion from natural language with high alignment to text and strong physical plausibility. Evaluations on the popular InterHuman Liang et al. (2024) and InterX Xu et al. (2024a) benchmarks demonstrate that our model significantly outperforms SOTA counterparts across all major metrics, including FID, R-Precision, and Multimodal Distance. Ablation studies confirm the value of our architectural decisions. Our main contributions include:

- We propose a disentangled hierarchical VAE which separates the latent representation of human-human interactions into three components: the individual motion components, and the global interaction component, enabling controllable and personalized generation.

- We propose a contrastive learning strategy over the global interaction latent $\mathbf{z}_o$ to enable prior-based modeling of interaction semantics and improved physical plausibility, especially for contact-sensitive regions.

- Our model is the lightest and fastest, and sets new SOTA performance on the InterHuman and InterX benchmarks on multiple metrics, demonstrating superior text-motion alignment and fidelity.

## 2    RELATED WORK

**Human Motion Generation.** Recent advances in human motion generation have been fueled by large-scale motion capture datasets and the emergence of generative models. Early methods relied on aligning latent spaces between text and motion using objectives like Kullback-Leibler divergence Guo et al. (2020; 2022b), or directly learning motion embeddings from paired descriptions Ahuja & Morency (2019). With the advent of Transformer models, autoregressive approaches such as T2M-GPT Zhang et al. (2023) and MotionGPT Jiang et al. (2024a) represent motion as discrete tokens and generate them sequentially. Despite producing coherent results, these methods often struggle with long-term dependencies and lack bidirectional modeling. To address these limitations, non-autoregressive models have gained popularity, including masked motion transformers such as MoMask Guo et al. (2024) and MMM Pinyoanuntapong et al. (2024), which leverage bidirectional context prediction via masked token reconstruction. In parallel, diffusion-based frameworks like MDM Tevet et al. (2023), MotionDiffuse Zhang et al. (2024), and FLAME Kim et al. (2023) have achieved promising results in generating temporally consistent and realistic motion. However, these methods are designed for single-person motion and struggle to model complex multi-agent interactions.

**Human-Human Interaction Generation.** Human-human interaction modeling extends the challenges of motion generation by requiring coherent inter-personal dynamics. Prior works can be broadly categorized into *reaction-based* and *joint generation* paradigms. Reaction models such as Chopin et al. (2024); Ghosh et al. (2024); Xu et al. (2024b); Ji et al. (2025) generate one agent's motion conditioned on another's actions, but often lack symmetry or generalization across diverse interaction types. Joint generation approaches like ComMDM Shafir et al. (2024) and RIG Tanaka & Fujiwara (2023) adopt diffusion-based architectures where two agents are jointly denoised with shared or cross-conditioned modules. InterGen Liang et al. (2024) extends this by using cooperative denoisers with mutual conditioning. Later methods Shan et al. (2024); Fan et al. (2024); Li et al. (2025) further scale to HHI by introducing global relational attention or structured priors. Ruiz-Ponce et al. (2024) realized the post effort to control interaction, and proposed in2IN by generating individual motions prior separately and then refining interaction with guided diffusion. Wang et al. (2025) noticed the problem of role-aware interaction and put up Role-Aware Scan and Localized Pattern Amplification to enforce the accuracy of interaction. Despite their success, these approaches still face limitations in fine-grained spatiotemporal modeling and high-quality coordination across different bodies, especially when interactions are diverse or weakly defined. Hence, Javed et al. (2025) proposed InterMask, a BERT-like Devlin et al. (2019) model which encodes the spatial and temporal representation in a discrete token space and generates the masked token gradually from scratch.

**Latent Diffusion for Structured Generation.** Latent diffusion models (LDMs) Rombach et al. (2021) have emerged as a scalable solution for high-dimensional generative tasks by performing the diffusion process in a compressed latent space, significantly reducing computational cost while maintaining expressivity. In single-person motion synthesis, latent diffusion has improved sampling efficiency and modeling capacity, as demonstrated by works like MLD, which uses a transformer-based VAE to compress long motion sequences in a compressed unified latent token. For HHI, Li et al. (2025) follows the architecture of MLD and compresses the two-person motions in a single latent representation. However, these methods typically operate in a flat latent space and do not capture the hierarchical or multi-scale nature of HHI motion patterns. To address this, our proposed method introduces a hierarchical latent diffusion architecture that models motion generation and interaction at multiple levels of abstraction. By integrating both local joint-level dynamics and global interaction-level context in a unified framework, we achieve improved realism, controllability, and diversity in HHI motion synthesis.

## 3    METHOD

### 3.1    DISENTANGLED HIERARCHICAL LATENT SPACE ENCODING

As shown in Fig. 2, we propose a Disentangled Hierarchical Variational Autoencoder (DHVAE) that disentangles motion in a global-individual manner via three latent variables: $\mathbf{z}_a$ and $\mathbf{z}_b$ for Persons A ($x_a$) and B ($x_b$), ensuring personalized motion details, and a joint latent $\mathbf{z}_o$ capturing their global semantics and interaction.

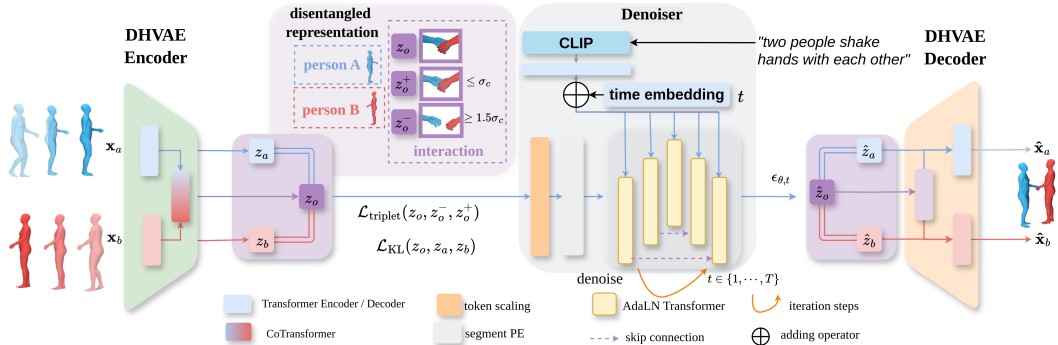

Figure 2: Architecture of our DHVAE to encode the structured latent representation $\mathbf{z}_o, \mathbf{z}_a, \mathbf{z}_b$. The global latent token $\mathbf{z}_o$ will learn an interaction plausible space via contrastive learning. The encoded structured representation will be passed into a skip-connected AdaLN Transformer to learn the denoise process.

**Encoding.** Individual motion features are extracted using Transformer encoders with learnable tokens $u_a, u_b$ (see Sec.6.14), producing $\mathbf{z}_a$ and $\mathbf{z}_b$ and individual temporal embeddings. A CoTransformer fuses these individual embeddings to model interactions. Each branch uses the other's output as key and value, with skip connections reducing query distortion. Outputs are concatenated with a global token $u_o$ and passed through an MLP to form $\mathbf{z}_o$, modeled as a Gaussian latent with learnable mean and variance.

**Decoding.** The global latent $\mathbf{z}_o$ is first decoded via a Transformer to obtain implicit interaction, then fed to two parallel Transformer decoders for Persons A and B. Each decoder attends to $\mathbf{z}_o$ through cross-attention, generating temporally synchronized and semantically coherent motion sequences.

**Objective Function.** Let $\mathbf{x} = [\mathbf{x}_a, \mathbf{x}_b]$ be the motion of the two persons. A conventional (flat) VAE like Li et al. (2025) would model it as:

$$\log p(\mathbf{x}) \geq \mathbb{E}_{q(\mathbf{z}|\mathbf{x})} \left[ \log p(\mathbf{x}|\mathbf{z}) \right] - D_{\mathrm{KL}} \left[ q(\mathbf{z}|\mathbf{x}) \| p(\mathbf{z}) \right], \tag{1}$$

While Equation 1 models the joint distribution of $\mathbf{x}_a$ and $\mathbf{x}_b$, it employs a single latent variable $\mathbf{z}$ and therefore does not explicitly separate agent-specific and shared semantic features or global interactions. In contrast, our DHVAE introduces structured latent variables $\mathbf{z}_a$, $\mathbf{z}_b$, and $\mathbf{z}_o$ to capture both private behaviors and shared dependencies:

$$\begin{aligned}
\log p(\mathbf{x}_a, \mathbf{x}_b) \geq \mathcal{L}_{\mathrm{ELBO}} = \mathbb{E}_{q(\mathbf{z}_a, \mathbf{z}_b, \mathbf{z}_o|\mathbf{x})} & \left[ \log p(\mathbf{x}_a|\mathbf{z}_o, \mathbf{z}_a) + \log p(\mathbf{x}_b|\mathbf{z}_o, \mathbf{z}_b) \right] \\
- D_{\mathrm{KL}} \left[ q(\mathbf{z}_a|\mathbf{x}_a) \| p(\mathbf{z}_a) \right] - D_{\mathrm{KL}} & \left[ q(\mathbf{z}_b|\mathbf{x}_b) \| p(\mathbf{z}_b) \right] - D_{\mathrm{KL}} \left[ q(\mathbf{z}_o|\mathbf{z}_a, \mathbf{z}_b) \| p(\mathbf{z}_o) \right],
\end{aligned} \tag{2}$$

where $p(\mathbf{x}_a|\mathbf{z}_o, \mathbf{z}_a)$ / $p(\mathbf{x}_b|\mathbf{z}_o, \mathbf{z}_b)$ denote individual likelihoods reconstructed from global and individual latents, $q(\mathbf{z}_a|\mathbf{x}_a)$ / $q(\mathbf{z}_b|\mathbf{x}_b)$ are individual posterior distributions, $q(\mathbf{z}_o|\mathbf{z}_a, \mathbf{z}_b)$ is the CoTransformer-encoded global interaction latent, and $p(\mathbf{z}_o)$, $p(\mathbf{z}_a)$, $p(\mathbf{z}_b)$ are Gaussian priors. This formulation captures causal or complementary inter-agent dynamics, enabling structured interaction modeling, one-to-many conditional generation, and fine-grained semantic control. In contrast, works like Li et al. (2025) model only $p(\mathbf{x}_a, \mathbf{x}_b|\mathbf{z})$, which overly compresses information and leads to large covariance between agents. While our hierarchical decoding manner reduces the covariance between components, increasing the capability of the denoising process (Please see Sec 6.6 for theoretical and visualization discussion).

**Interaction Contrastive Learning.** Prior methods, such as InterGen and InterLDM, impose a penalty on the pairwise distance between agents to encourage interaction-aware behavior. However, this approach assumes a fixed spatial proximity between agents, overlooking the significant variability between contact and non-contact interactions. As a result, such methods tend to overfit to specific motion patterns and fail to generalize across diverse interaction types. This often leads to implausible behaviors such as penetration or unnatural detachment.

To address this limitation, we propose a contrastive learning objective over the global interaction latent variable $\mathbf{z}_o$, which encodes the shared context between agents. By constructing positive and negative motion pairs based on semantic and physical plausibility, our method encourages $\mathbf{z}_o$ to capture meaningful interaction structures. This not only enhances generalization across different interaction scenarios but also improves the physical realism and coherence of the generated motions.

The core idea is summarized in Algorithm 1. For each motion pair $(\mathbf{x}_a, \mathbf{x}_b)$, we first check for physical contact by computing the overlap between the voxelized human mesh pairs. If contact exists, we create a positive sample $\mathbf{x}_b^+$ by applying a small ground-plane translation within $\pm\sigma_c$; otherwise, we allow a slightly larger range $\pm\sigma_u$. A negative sample $\mathbf{x}_b^-$ is generated by a larger shift from a two-tailed truncated Gaussian, ensuring spatial inconsistency. Passing these perturbed pairs into DHVAE yields interaction latents $\mathbf{z}_o^+$ and $\mathbf{z}_o^-$. A triplet margin loss then enforces $\mathbf{z}_o$ to be closer to $\mathbf{z}_o^+$ than $\mathbf{z}_o^-$ by margin $m$, encouraging sensitivity to spatial plausibility. To preserve joint information, we add a joint-position penalty. The overall objective is:

$$\mathcal{L}_{\text{DHVAE}} = \mathcal{L}_{\text{ELBO}} + \lambda_{\text{joint}}\mathcal{L}_{\text{joint}} + \lambda_{\text{triplet}}\mathcal{L}_{\text{triplet}}, \tag{3}$$

where $\mathcal{L}_{\text{joint}}$ is the L1 loss on joint positions. For InterHuman, we use global positions, while for InterX, we derive joints via the SMPLX forward kinematics.

---

**Algorithm 1** Contrastive Learning for Interaction Latent $\mathbf{z}_o$.

---

**Require:** Input motion pair $(\mathbf{x}_a, \mathbf{x}_b)$
  1: Compute global latent: $\mathbf{z}_o = \text{DHVAE}(\mathbf{x}_a, \mathbf{x}_b)$
  2: **if** is_contact$(\mathbf{x}_a, \mathbf{x}_b)$ **then**
  3:     $\mathbf{x}_b^+ \leftarrow \text{Translate}(\mathbf{x}_b, \Delta \sim \text{TruncNorm}(0, \sigma = \sigma_c))$ {$\sigma_c \approx 5\text{cm, small jitter}$}
  4: **else**
  5:     $\mathbf{x}_b^+ \leftarrow \text{Translate}(\mathbf{x}_b, \Delta \sim \text{TruncNorm}(0, \sigma = \sigma_u))$ {$\sigma_u \approx 30\text{cm, looser non-contact jitter}$}
  6: **end if**
  7: $\mathbf{x}_b^- \leftarrow \text{Translate}(\mathbf{x}_b, \Delta \sim \text{TwoTailed}(\pm 1.5\sigma, \pm 3\sigma))$ {$\Delta \approx 45\text{–}90\text{cm, implausible shift}$}
  8: $\mathbf{z}_o^+ = \text{DHVAE}(\mathbf{x}_a, \mathbf{x}_b^+), \quad \mathbf{z}_o^- = \text{DHVAE}(\mathbf{x}_a, \mathbf{x}_b^-)$
  9: Compute contrastive loss:

$$\mathcal{L}_{\text{triplet}} = \max(0, d(\mathbf{z}_o, \mathbf{z}_o^+) - d(\mathbf{z}_o, \mathbf{z}_o^-) + m)$$

---

## 3.2 DIFFUSION OF HIERARCHICAL LATENT

DHVAE provides a disentangled latent representation $\{\mathbf{z}_o, \mathbf{z}_a, \mathbf{z}_b\}$ for global interaction and individual motions. To generate realistic and diverse sequences, we perform latent diffusion Rombach et al. (2021) using DDIM Song et al. (2021), which enables efficient non-Markovian sampling without loss of quality. The forward process gradually adds Gaussian noise,

$$q(\mathbf{z}_t|\mathbf{z}_{t-1}) = \mathcal{N}(\mathbf{z}_t; \sqrt{1-\beta_t}\mathbf{z}_{t-1}, \beta_t\mathbf{I}), \tag{4}$$

with reparameterization

$$\mathbf{z}_t = \sqrt{\bar{\alpha}_t}\mathbf{z}_0 + \sqrt{1-\bar{\alpha}_t}\epsilon, \quad \epsilon \sim \mathcal{N}(0, \mathbf{I}), \tag{5}$$

where $\bar{\alpha}_t = \prod_{s=1}^{t}(1-\beta_s)$. A Transformer-based denoiser $\epsilon_\theta(\mathbf{z}_t, t, c)$ reconstructs all latent components conditioned on text $c$, and the DHVAE decoder synchronizes them back into motion sequences. To stabilize training, we apply token scaling to balance contributions of $\mathbf{z}_o$, $\mathbf{z}_a$, and $\mathbf{z}_b$. Moreover, since these components have different value ranges, we normalize $\mathbf{z}_a$ and $\mathbf{z}_b$ by a scale factor $s_l$ so that $\mathbf{z} = \{\mathbf{z}_o, \mathbf{z}_a/s_l, \mathbf{z}_b/s_l\}$ lies in a comparable range.

Besides, to learn the hierarchical order, we employ a SiLU-based MLP combined with an embedding layer instead of traditional sinusoidal embedding. InterGen Liang et al. (2024) first introduced AdaLN for HHI and later works followed. The AdaLN used in InterGen and in2IN has a two-parameter setting with only a scale $\beta$ and a shift $\mu$, whereas we follow InterMask, which has an AdaLN-zero style three-parameter setting. While stacking multiple Transformer layers enables the model to capture multi-level information, deeper stacks alone often lead to vanishing gradients, especially in the denoising process. To improve the capacity and stability of the denoising process, we

adopt a U-Net-like architecture Ronneberger et al. (2015) within the AdaLN Transformer-based denoiser. This design introduces skip connections between the opposite layers in the denoiser, enabling the reuse of low-level latent features in shallow layers.

**Classifier-Free Guidance.** To enhance sample diversity and controllability, we incorporate classifier-free guidance (CFG) Ho & Salimans (2021) during denoising. The model is jointly trained with and without textual conditions. During inference, the denoised output is guided by interpolating between the unconditional and conditional predictions with a guidance strength $\omega$:

$$\epsilon_{\text{guided}} = (1 + \omega) \cdot \epsilon_\theta(\mathbf{z}_t, t, c) - \omega \cdot \epsilon_\theta(\mathbf{z}_t, t), \tag{6}$$

## 4 EXPERIMENTS

**Datasets.** We perform comparative evaluations on two popular benchmarks for text-conditioned human interaction generation: *InterHuman* Liang et al. (2024) and *InterX* Xu et al. (2024a). *InterHuman* adopts the AMASS Mahmood et al. (2019) skeleton format with 22 joints, including the root joint. Each joint but for the root is described by the tuple $\{\mathbf{p}_g, \mathbf{v}_g, \mathbf{r}_{6d}\}$, where $\mathbf{p}_g \in \mathbb{R}^3$ is the global position, $\mathbf{v}_g \in \mathbb{R}^3$ is the global velocity, and $\mathbf{r}_{6d} \in \mathbb{R}^6$ represents the local 6D rotation. This results in a motion representation tensor $\mathbf{m}_p \in \mathbb{R}^{N \times 262}$. *InterX* follows the SMPL-X Pavlakos et al. (2019) format comprising rotations of 55 joints, including the main body, hands, and face joints, along with the root orientation. Each joint and root orientation is represented by $\mathbf{r}_{6d}$, and the root translation by $\mathbf{p}_g$, with additional root velocity $\mathbf{v}_g$ added to the representation. This results in $\mathbf{m}_p \in \mathbb{R}^{N \times 56 \times 6}$. We retain the native skeleton representations of each dataset to demonstrate the compatibility of our framework with diverse pose representations and joint structures.

**Evaluation Metrics.** To comprehensively evaluate the performance of our model, we adopt a suite of feature-space metrics by Liang et al. (2024). To assess the realism and fidelity of generated interactions, we compute the Fréchet Inception Distance (FID) and between the feature distributions of generated and ground-truth motions and their Diversity. To evaluate the semantic alignment between text prompts and generated motions, we use R-Precision and Multimodal Distance (MMDist), which measure the consistency between text input and generated motions. To assess generative quality beyond accuracy, we report Multimodality (MModality), quantifying the model's ability to produce multiple motions for the same textual description.

**Baselines.** We compare against all SOTA HHI generation methods, including InterGen Liang et al. (2024), MoMat-MoGen Cai et al. (2024), InterMask Javed et al. (2025), and in2IN Ruiz-Ponce et al. (2024) on the InterHuman dataset. For InterX, we follow the evaluation setup of Javed et al. (2025) and report results for T2M Zhang et al. (2023), MDM Tevet et al. (2023), ComMDM Shafir et al. (2024), InterGen, and InterMask. Since in2IN is trained on HumanML3D-style motion representations, it cannot be applied to the SMPLX-based InterX dataset. In addition, we implement an MLD-based variant as a latent diffusion baseline for both datasets. We also include TIMotion Wang et al. (2025) for comparison. However, since only the InterHuman variant is officially released, we run their provided code and pretrained weights to reproduce results, and for InterX, we directly use their claimed results.

**Implementation Details.** We use a latent space of $1 \times 256$ for $\mathbf{z}_o, \mathbf{z}_a, \mathbf{z}_b$ for the InterHuman and $1 \times 336$ for the InterX due to different representation dimensions (InterHuman is 262 and InterX is 336). For the DHVAE encoder, we employed two sets of Transformer Encoders of 4 layers, and a CoTransformer of 3 layers with skip connection, which have a hidden dimension of 1024, a head number of 4, and a dropout ratio of 0.1 for InterHuman; and a hidden dimension of 1344 for the InterX Dataset. For the DHVAE decoder, all three decoders share the same parameter settings as the corresponding encoders. For the denoiser, we use 13 layers of AdaLN Transformer, and other settings follow the Transformer above. In the reverse diffusion process, we set the $\beta_0 = 0.00085$ and $\beta_T = 0.012$, the diffusion time steps to be $T = 1000$, and the inference time step to be 50. We set the unconditional ratio to be 0.1 and use the CFG strength of 3.5/3.0 for the InterHuman/InterX datasets. We trained the DHVAE for 2000/1200 epochs for InterHuman/InterX and 1600/1000 epochs for the denoiser. All experiments were conducted on a single NVIDIA H100 GPU.

### 4.1 QUANTITATIVE RESULTS

We conduct comprehensive evaluations on both the benchmarks, where our model achieves SOTA performance across all major metrics. As shown in Table 1, our method consistently outperforms

| InterHuman | | | | | | |
|---|---|---|---|---|---|---|
| Model | R-Prec@1 ↑ | R-Prec@2 ↑ | R-Prec@3 ↑ | FID ↓ | MM Dist ↓ | Diversity → | MModality ↑ |
| Ground Truth | $0.452^{\pm.008}$ | $0.610^{\pm.009}$ | $0.701^{\pm.008}$ | $0.273^{\pm.007}$ | $3.755^{\pm.008}$ | $7.948^{\pm.064}$ | - |
| InterGen Liang et al. (2024) | $0.371^{\pm.010}$ | $0.515^{\pm.012}$ | $0.624^{\pm.010}$ | $5.918^{\pm.079}$ | $5.108^{\pm.014}$ | $7.387^{\pm.029}$ | $\mathbf{2.141}^{\pm.063}$ |
| MLD* Chen et al. (2023) | $0.392^{\pm.005}$ | $0.533^{\pm.005}$ | $0.612^{\pm.004}$ | $6.158^{\pm.082}$ | $3.817^{\pm.003}$ | $7.785^{\pm.048}$ | $1.236^{\pm.035}$ |
| MoMat-MoGen Cai et al. (2024) | $0.449^{\pm.004}$ | $0.591^{\pm.003}$ | $0.666^{\pm.004}$ | $5.674^{\pm.120}$ | $3.790^{\pm.001}$ | $8.021^{\pm.035}$ | $1.295^{\pm.023}$ |
| in2IN Ruiz-Ponce et al. (2024) | $0.455^{\pm.004}$ | $0.611^{\pm.008}$ | $0.687^{\pm.009}$ | $5.177^{\pm.120}$ | $3.790^{\pm.002}$ | $7.940^{\pm.047}$ | $1.061^{\pm.038}$ |
| InterMask Javed et al. (2025) | $0.449^{\pm.005}$ | $0.599^{\pm.005}$ | $0.681^{\pm.004}$ | $\underline{5.153}^{\pm.061}$ | $3.790^{\pm.002}$ | $\mathbf{7.944}^{\pm.033}$ | $\underline{1.737}^{\pm.020}$ |
| TIMotion* Wang et al. (2025) | $\underline{0.485}^{\pm.007}$ | $\underline{0.635}^{\pm.005}$ | $\underline{0.712}^{\pm.005}$ | $5.600^{\pm.106}$ | $\underline{3.779}^{\pm.002}$ | $7.964^{\pm.029}$ | $0.952^{\pm.023}$ |
| **Ours** | $\mathbf{0.496}^{\pm.004}$ | $\mathbf{0.647}^{\pm.006}$ | $\mathbf{0.720}^{\pm.005}$ | $\mathbf{5.015}^{\pm.085}$ | $\mathbf{3.772}^{\pm.002}$ | $\underline{7.952}^{\pm.045}$ | $0.804^{\pm.030}$ |
| InterX | | | | | | |
| Model | R-Prec@1 ↑ | R-Prec@2 ↑ | R-Prec@3 ↑ | FID ↓ | MM Dist ↓ | Diversity → | MModality ↑ |
| Ground Truth | $0.429^{\pm.004}$ | $0.626^{\pm.003}$ | $0.736^{\pm.003}$ | $0.002^{\pm.000}$ | $3.536^{\pm.013}$ | $9.734^{\pm.078}$ | - |
| T2M Zhang et al. (2023) | $0.184^{\pm.010}$ | $0.298^{\pm.010}$ | $0.396^{\pm.005}$ | $5.481^{\pm.382}$ | $9.576^{\pm.006}$ | $2.771^{\pm.151}$ | $2.761^{\pm.042}$ |
| MDM Tevet et al. (2023) | $0.203^{\pm.020}$ | $0.329^{\pm.007}$ | $0.426^{\pm.005}$ | $23.701^{\pm.057}$ | $9.548^{\pm.017}$ | $5.856^{\pm.077}$ | $\underline{3.490}^{\pm.061}$ |
| ComMDM Shafir et al. (2024) | $0.090^{\pm.009}$ | $0.165^{\pm.004}$ | $0.236^{\pm.003}$ | $29.266^{\pm.067}$ | $6.870^{\pm.017}$ | $4.734^{\pm.067}$ | $0.771^{\pm.053}$ |
| InterGen Liang et al. (2024) | $0.207^{\pm.020}$ | $0.335^{\pm.002}$ | $0.429^{\pm.004}$ | $5.207^{\pm.216}$ | $8.504^{\pm.057}$ | $7.788^{\pm.208}$ | $\mathbf{3.686}^{\pm.052}$ |
| MLD* Chen et al. (2023) | $0.386^{\pm.006}$ | $0.577^{\pm.005}$ | $0.678^{\pm.005}$ | $1.295^{\pm.038}$ | $4.056^{\pm.033}$ | $9.008^{\pm.102}$ | $2.375^{\pm.047}$ |
| InterMask Javed et al. (2025) | $0.403^{\pm.005}$ | $0.595^{\pm.006}$ | $0.705^{\pm.005}$ | $0.399^{\pm.013}$ | $\underline{3.705}^{\pm.017}$ | $9.046^{\pm.073}$ | $2.261^{\pm.081}$ |
| TIMotion Wang et al. (2025) | $\underline{0.412}^{\pm.004}$ | $\underline{0.601}^{\pm.004}$ | $\underline{0.714}^{\pm.003}$ | $\underline{0.385}^{\pm.022}$ | $3.706^{\pm.015}$ | $\underline{9.191}^{\pm.092}$ | $2.437^{\pm.069}$ |
| **Ours** | $\mathbf{0.442}^{\pm.005}$ | $\mathbf{0.638}^{\pm.005}$ | $\mathbf{0.745}^{\pm.004}$ | $\mathbf{0.339}^{\pm.015}$ | $\mathbf{3.604}^{\pm.020}$ | $\mathbf{9.378}^{\pm.065}$ | $2.505^{\pm.047}$ |

Table 1: Comparisons on InterHuman and InterX datasets. The best results are in **bold**, and the second-best are underlined. Methods with * are implemented by us. All results are run 20 times. For a fair comparison, we set the latent size of MLD to be the same as ours, i.e. $3 \times 256$.

previous approaches by a significant margin with the lowest FID, MMDist, and the highest R-precisions. These results demonstrate our model's efficacy in producing realistic and coherent HHI sequences.

As a two-step generation pipeline, encoder reconstruction sets an upper bound on the model's fidelity. To evaluate the reconstruction quality of our DHVAE, we compare it against the VAEs of two other two-step methods: InterMask (2D-VQ-VAE) and our implementation of MLD, as previously shown in Fig. 1. As shown in Table 2, DHVAE achieves the best reconstruction performance across reconstruction FID (rFID), MPJPE (in meters), and L1 loss on Z-normalized features, which suggests that our disentangled prior enables compact and structured representation.

Besides, we evaluate computational efficiency in terms of model size and average inference time per sentence (AITS, measured in seconds and excluding the pretrained CLIP text encoder). As shown in Table 3, our model outperforms the SOTA methods TIMotion and InterMask, achieving both the smallest parameter footprint and the fastest inference speed. This highlights the lightweight yet efficient design of our approach. To the best of our knowledge, DHVAE is the first work since Inter-Mask to achieve consistent improvements across all core metrics on both InterHuman and InterX, thereby establishing a new state-of-the-art baseline for text-conditioned HHI generation.

Table 2: Reconstruction results for prior-based and SOTA models, best results in bold.

| Task | Model | rFID ↓ | MPJPE ↓ | L1 Loss ↓ |
|---|---|---|---|---|
| | MLD-VAE | 1.011 | 0.089 | 0.256 |
| Recon | 2D-VQ-VAE | 0.970 | 0.129 | 0.276 |
| | DHVAE (ours) | **0.503** | **0.055** | **0.218** |

Table 3: Computational cost of models including latency and size

| Model | AITS ↓ | Size |
|---|---|---|
| InterMask | 1.021 | 74M |
| TIMotion | 1.472 | 77M |
| **DHVAE (ours)** | **0.454** | **56M** |

Finally, we evaluate the physical plausibility of generated motions by computing both penetration statistics and contact quality. Following prior work Jiang et al. (2024b), we voxelize the SMPL-X Pavlakos et al. (2019) mesh at a resolution of 2 cm and measure the degree of interpenetration based on voxel overlap. Specifically, we report the *Penetration Volume* (PV), defined as the average number of overlapping voxels across all generated sequences. To capture different aspects of penetration, we further introduce two complementary metrics: the *Penetration Frequency Ratio* (PFR), which measures the proportion of sequences that contain any penetration event, and the *Penetration Duration Ratio* (PDR), which quantifies the temporal fraction of penetration within each sequence. In addition to penetration measures, we compute the *contact ratio* for sequences with annotated con-

tact. To ensure robustness, we apply voxel dilation to both interacting meshes and treat cases where the maximum overlapping voxel count exceeds the volume of a $6\,\mathrm{cm}$ cube (i.e., $216\,\mathrm{ml}$) as severe penetrations, which are subsequently excluded from being considered valid contact. This prevents false positives caused by deep or unrealistic intersections and ensures that the contact score reflects physically meaningful interactions.

The results in Table 4 suggest that our method achieves the lowest penetration score and the highest contact ratio among MLD, InterMask, and TIMotion.

| Model | PV $\downarrow$ | PFR $\downarrow$ | PDR $\downarrow$ | Contact $\uparrow$ |
|---|---|---|---|---|
| MLD* | 0.503 | 0.108 | 0.175 | 0.427 |
| InterMask | 0.873 | 0.149 | 0.243 | 0.349 |
| TIMotion | 0.485 | 0.122 | 0.104 | 0.466 |
| Ours w/o triplet | 0.446 | 0.107 | 0.102 | 0.445 |
| Ours | **0.390** | **0.064** | **0.087** | **0.581** |

Table 4: Penetration metrics and contact ratio for state-of-the-art models.

## 4.2 QUALITATIVE RESULTS

We compare our method with the SOTA baseline InterMask. As no official InterMask implementation exists for InterX and space is limited, we show visual results only on InterHuman (Fig. 3). The results with TIMotion are presented in Fig. 7. InterMask encodes individuals independently in a discrete token space without a global interaction latent, often yielding implausible motions at contact points, body interpenetrations (e.g., "*hug the seated person*"), or failures to follow complex prompts (e.g., "*one standing up, the other reaching out a hand*"). In contrast, our method produces more physically plausible and semantically accurate interactions, such as successfully aligning hands for "*greet each other by shaking hands.*"

## 4.3 ABLATION STUDY

**DHVAE.** We compare our proposed DHVAE with a baseline VAE based on the MLD framework Chen et al. (2023), which uses a unified latent representation equipped with a skip-Transformer. For a fair comparison, we align the major architectural configurations, including latent dimensionality ($3 \times 256$ for MLD-VAE) and the denoiser used to model the reverse diffusion process. From Tables 1 and 5, we can observe that by replacing the VAE with the MLD-VAE, there is a significant degradation, suggesting the disentangled latent space is better than a uniform flattened

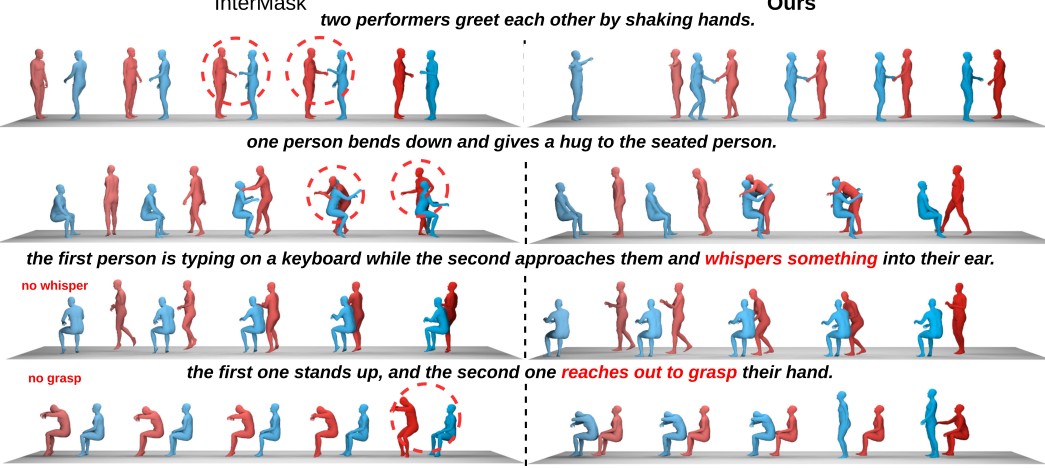

Figure 3: Qualitative Comparison with InterMask Javed et al. (2025) on InterHuman Dataset, indicating superior text alignment, fidelity, and physical plausibility. The body meshes are arranged sequentially from left to right, with colors progressing from light to dark.

| DHVAE | | | | Denoiser | | | Metrics | | |
|---|---|---|---|---|---|---|---|---|---|
| Triplet Loss | $\mathbf{z}_o$ | CoTRM | MLD-VAE | SPE | TokenScale | Skip | rFID ↓ | gFID ↓ | RP@1 ↑ |
| - | ✓ | ✓ | - | ✓ | ✓ | ✓ | **0.499** | 5.063 | 0.492 |
| ✓ | - | ✓ | - | ✓ | ✓ | ✓ | 0.669 | 5.886 | 0.468 |
| - | - | ✓ | - | ✓ | ✓ | ✓ | 0.667 | 6.005 | 0.463 |
| ✓ | ✓ | - | - | ✓ | ✓ | ✓ | 0.632 | 5.593 | 0.476 |
| - | - | - | ✓ | ✓ | ✓ | ✓ | 1.024 | 5.694 | 0.457 |
| ✓ | - | - | ✓ | ✓ | ✓ | ✓ | 1.089 | 5.603 | 0.462 |
| ✓ | ✓ | ✓ | - | - | ✓ | ✓ | - | 7.452 | 0.423 |
| ✓ | ✓ | ✓ | - | ✓ | - | ✓ | - | 6.531 | 0.413 |
| ✓ | ✓ | ✓ | - | ✓ | ✓ | - | - | 5.343 | 0.468 |
| ✓ | ✓ | ✓ | - | ✓ | ✓ | ✓ | 0.503 | **5.015** | **0.496** |

Table 5: Ablation study of DHVAE and Denoiser components. ✓ indicates the component is used. For a fair comparison with MLD, we set the latent size of MLD-VAE to be $3 \times 256$

one. We further ablate the effect of interaction contrastive learning by removing the triplet loss. This leads to a slight drop in reconstruction accuracy but improves generation quality. Although the numerical metrics show no obvious advantage, qualitative results in Sec.6.5 and physical evaluations in Table 4 reveal that the triplet loss improves the physical plausibility. Additionally, we evaluate the impact of two key components in DHVAE: (1) the CoTransformer, and (2) the global latent variable $\mathbf{z}_o$. Specifically, we consider two variants: replacing the CoTransformer with a 3-layer MLP with residual connections, and removing the global latent $\mathbf{z}_o$, generating motions solely from $\mathbf{z}_a$ and $\mathbf{z}_b$ (maintaining CoTransformer). As shown in line 2 & 4, both modifications result in substantial performance drops in reconstruction and generation, confirming the importance of hierarchical latent modeling and effective context aggregation via the CoTransformer. **Denoiser.** We ablate three architectural design choices for the denoiser: (1) SPE, (2) token scaling, and (3) skip connections. To assess the role of SPE, we replace it with standard sinusoidal positional embeddings. As shown in Table 5, removing either SPE or token scaling leads to significant performance degradation, highlighting the importance of encoding structural segmentation and maintaining scale consistency across latent components. While omitting skip connections has a smaller impact, it still provides measurable benefits, particularly in improving convergence during the reverse diffusion process.

## 5 CONCLUSION

We proposed a Disentangled Hierarchical VAE-based latent diffusion framework for text-conditioned HHI generation. By disentangling global interaction and individual motions and employing a CoTransformer module, our approach addresses the limitations of existing flat latent representations. The denoiser integrated with skip-connected AdaLN-Transformer, segment positional embedding, and token scaling enables high-quality and controllable motion generation. Experiments on two challenging benchmarks demonstrate that our method achieves new state-of-the-art in realism and semantic alignment. Our approach offers a robust and extensible framework for HHI motion generation, with promising implications for animation, and human-robot collaboration. Future work could explore incorporating social cues, expanding to more than two agents, and extending to 3D avatars or embodied simulation environments.

## ACKNOWLEDGEMENTS

This research was supported by the Australian Research Council (ARC) Discovery Project DP240101926. We gratefully acknowledge this support.

## ETHICS STATEMENT

Our work follows the ICLR Code of Ethics, which emphasizes contributing to society, minimizing harm, respecting privacy, and upholding high standards of scientific integrity. The proposed method does not involve direct human subjects, sensitive personal data, or identifiable information. All experiments are conducted on publicly available benchmark datasets under their respective licenses. We ensure transparency and reproducibility by providing detailed methodology and acknowledging all contributions.

We believe our research benefits the community by advancing motion representation learning in a socially responsible way. It avoids discrimination, respects fairness and inclusivity, and does not pose foreseeable risks to health, safety, or the natural environment. Should any unintended negative consequences arise, we are committed to mitigating them in accordance with ethical best practices.

## REPRODUCIBILITY STATEMENT

We have taken several steps to ensure the reproducibility of our work. All model architectures, training procedures, and hyperparameter settings are described in detail in Sections 6.14 and Section 4, with additional implementation details and ablation studies provided in the Appendix. The datasets used are publicly available, and all preprocessing steps are carefully documented in the supplementary materials. To further facilitate reproducibility, we will release the complete source code, pretrained models, and data processing scripts upon acceptance. This will allow other researchers to fully replicate and build upon our results.

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

## 6 APPENDIX

### 6.1 USER STUDY

We conducted a user study at a moderate scale, distributing 30 surveys and receiving 18 valid responses. In the study, we compared our results with InterMask and TIMotion across multiple motion categories using the InterHuman test set. The reviewers were asked to select the motion sequences with the highest quality—with all model names anonymized during evaluation. As illustrated in Fig. 4, our method consistently achieved the highest preference ratings compared to the other two approaches.

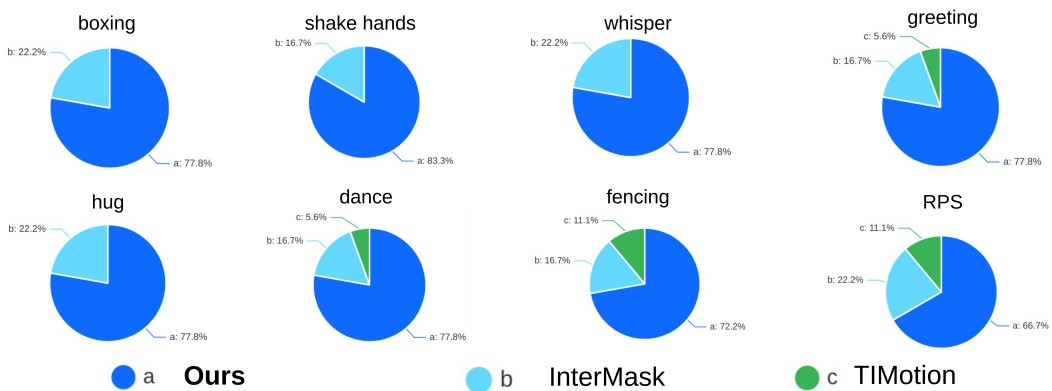

Figure 4: User Study of DHVAE compared to InterMask and TIMotion

### 6.2 CLASSIFIER FREE GUIDANCE

We investigate the impact of classifier-free guidance (CFG) on generation quality by varying the guidance scale from 0 to 10 in increments of 0.5. For each setting, we repeat the sampling process 20 times to account for randomness and report the average results. We compare our method against the InterMask baseline and highlight the regions where our approach surpasses the baseline using **gray** shading. The evaluation results for the InterHuman dataset are presented in Fig. 5, and the corresponding results for the InterX dataset are shown in Fig. 6.

This analysis provides insight into how different guidance strengths influence various metrics, including FID, R-Precision, and MM-Distance, thereby offering practical recommendations for optimal CFG values.

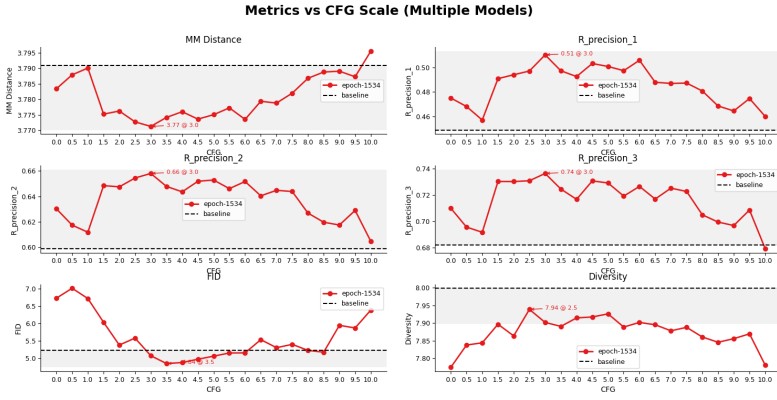

Figure 5: Metrics along the change along classifier-free-guidance scale on InterHuman dataset

## 6.3 QUANTITATIVE COMPARISON WITH INTERLMD

We consider InterLDM Li et al. (2025) to be the first application of latent diffusion models to the HHI task. However, we observed significant inconsistencies in its reported results—particularly an abnormally low MM-Distance—when compared with other state-of-the-art methods. Due to these inconsistencies and the inability to reproduce their results, we have chosen not to include InterLDM in the main comparison table (Table 1) to avoid introducing potentially misleading conclusions. Theoretically, the metrics of R-precision and Multimodal Distance share the **same embedding feature**, hence they are correlated, i.e., the higher the R-precision is, the lower the Multimodal Distance is. Therefore, we infer that their MMDistance can not match ours based on their R-precisions.

In our attempt to fairly benchmark their method, we noted that the authors did not release official code, and our direct inquiries for clarification received no clear response or suggestions. Additionally, InterLDM did not conduct evaluations on the InterX Xu et al. (2024a) benchmark, further limiting the completeness of the comparison. Nevertheless, for transparency and completeness, we include a side-by-side comparison with their reported results in the appendix. We emphasize that, except for the results on the questionnaire **MM-Distance**, our method outperforms InterLDM across all other reported metrics. This reinforces the effectiveness and robustness of our approach while addressing the ambiguity in their reported numbers.

| Dataset | Model | R-Prec@1 ↑ | R-Prec@2 ↑ | R-Prec@3 ↑ | FID ↓ | MM Dist ↓ | Diversity → |
|---|---|---|---|---|---|---|---|
| | Ground Truth | $0.452^{\pm0.008}$ | $0.610^{\pm0.007}$ | $0.701^{\pm0.008}$ | $0.273^{\pm0.007}$ | $3.755^{\pm0.008}$ | $7.948^{\pm0.064}$ |
| | InterGen Liang et al. (2024) | $0.371^{\pm.010}$ | $0.515^{\pm.012}$ | $0.624^{\pm.010}$ | $5.918^{\pm.079}$ | $5.108^{\pm.014}$ | $7.387^{\pm.029}$ |
| | MLD * Chen et al. (2023) | $0.392^{\pm.005}$ | $0.533^{\pm.005}$ | $0.612^{\pm.004}$ | $6.158^{\pm.082}$ | $3.817^{\pm.003}$ | $7.785^{\pm.048}$ |
| InterHuman | InterLDM | $0.427^{\pm.004}$ | $0.559^{\pm.050}$ | $0.638^{\pm.004}$ | $5.619^{\pm.091}$ | $1.862^{\pm.007}$ | $7.888^{\pm.041}$ |
| | MoMat-MoGen Cai et al. (2024) | $0.449^{\pm.004}$ | $0.591^{\pm.003}$ | $0.666^{\pm.004}$ | $5.674^{\pm.120}$ | $\underline{3.790}^{\pm.001}$ | $8.021^{\pm.035}$ |
| | InterMask Javed et al. (2025) | $0.449^{\pm.004}$ | $0.599^{\pm.005}$ | $0.681^{\pm.004}$ | $\underline{5.153}^{\pm.061}$ | $3.790^{\pm.002}$ | $\mathbf{7.944}^{\pm.033}$ |
| | in2IN Ruiz-Ponce et al. (2024) | $\underline{0.455}^{\pm.004}$ | $\underline{0.611}^{\pm.008}$ | $\underline{0.687}^{\pm.009}$ | $5.177^{\pm.120}$ | $3.790^{\pm.002}$ | $7.940^{\pm.047}$ |
| | **Ours** | $\mathbf{0.496}^{\pm.004}$ | $\mathbf{0.647}^{\pm.006}$ | $\mathbf{0.720}^{\pm.005}$ | $\mathbf{5.015}^{\pm.085}$ | $\mathbf{3.772}^{\pm.002}$ | $\underline{7.952}^{\pm.045}$ |

Table 6: Comparison results on InterHuman including **InterLDM**. The best results are marked with **bold** font, and the second-best results are marked with underline. The group with * is implemented by us. To avoid coincidence, each metric is repeated 20 times, and a standard deviation is provided.

## 6.4 QUALITATIVE COMPARISON WITH TIMOTION

As shown in Fig. 7, we compared the generated sequences with TIMotion with the same prompts, finding that TIMotion has serious artifacts, shaking, and what's more, **position shifting** between agents (marked with blue dash in the figure), which happened to most of the current works Liang et al. (2024); Tanaka & Fujiwara (2023). Besides, implausible contacts also happen, as marked in red circle. Please see the supplementary.mp4 file to see the actual animations.

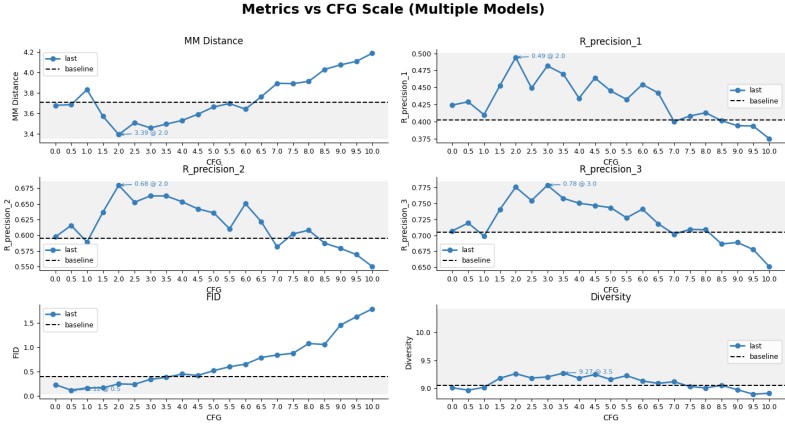

Figure 6: Metrics along the change along classifier-free-guidance scale on InterX dataset

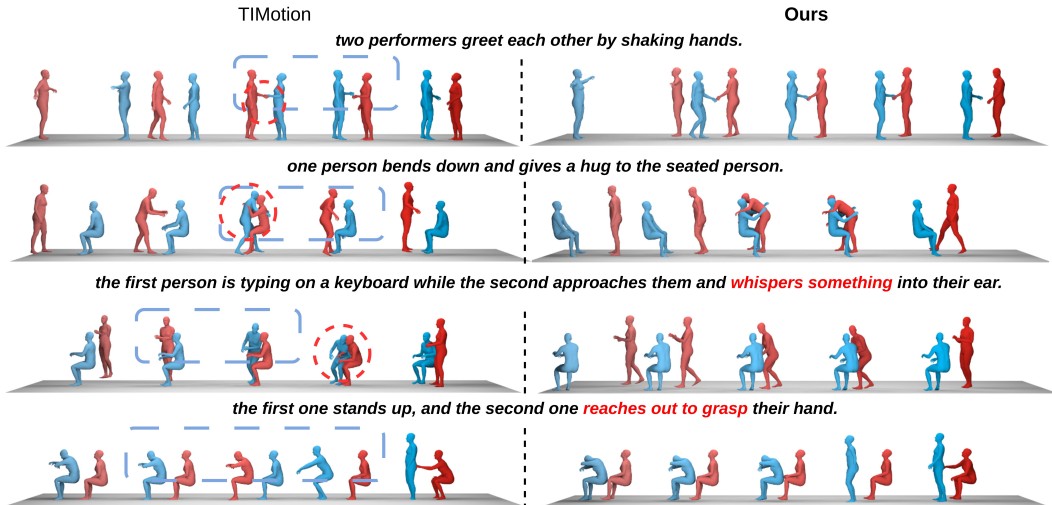

Figure 7: Comparison with TIMotion on InterHuman

### 6.5 VISUALIZATION OF ABLATION STUDY ON CONTRASTIVE LEARNING

In the main text, we have shown that contrastive learning improves the quantitative results. In this section, we present visual cases that, without the Contrastive Learning, part of the generated HHI sequences may not be semantic or physically plausible. As shown in the Fig. 9, for given prompts like "two people shake their hands", without the triplet loss, the hands do not come into contact closely,

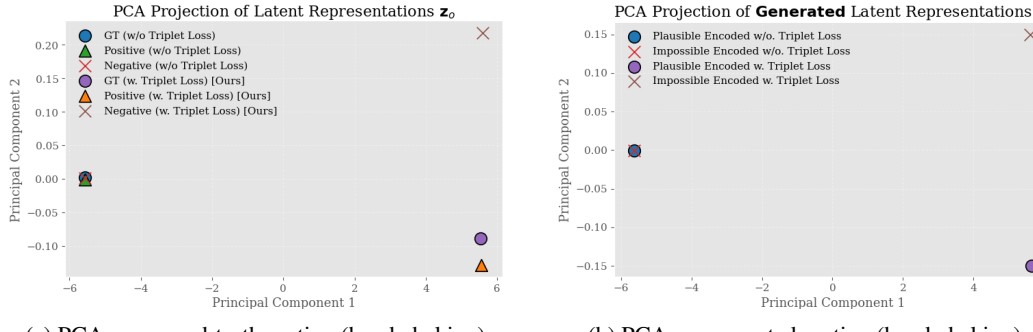

(a) PCA on ground-truth motion (hand-shaking).  (b) PCA on generated motion (hand-shaking).

Figure 8: PCA projections of the interaction latent $\mathbf{z}_o$ in the *hand-shaking* scenario. (a) shows encodings of ground-truth motion triplets; (b) visualizes $\mathbf{z}_o$ of generated motions from prompts. Both compare DHVAE with and without contrastive learning.

Based on the previous analysis, we visualize two PCA projections of the interaction latent $\mathbf{z}_o$ in the context of the *hand-shaking* task. Figure 8a illustrates the PCA embedding of the ground-truth motion, where we show the anchor, positive, and negative samples, encoded by DHVAE with and without contrastive learning. Figure 8b presents the encoded $\mathbf{z}_o$ from the generated motion sequences given the prompt "hand-shaking," again comparing models with and without contrastive learning. Our DHVAE model with contrastive learning clearly separates positive and negative samples in the latent space. This separation is consistent across both reconstruction from ground-truth motions and generation from language prompts. The structured latent space induced by contrastive supervision demonstrates not only improved representation learning but also contributes to more plausible and controllable generation quality. For detailed animations, please refer to the supplementary.mp4 file.

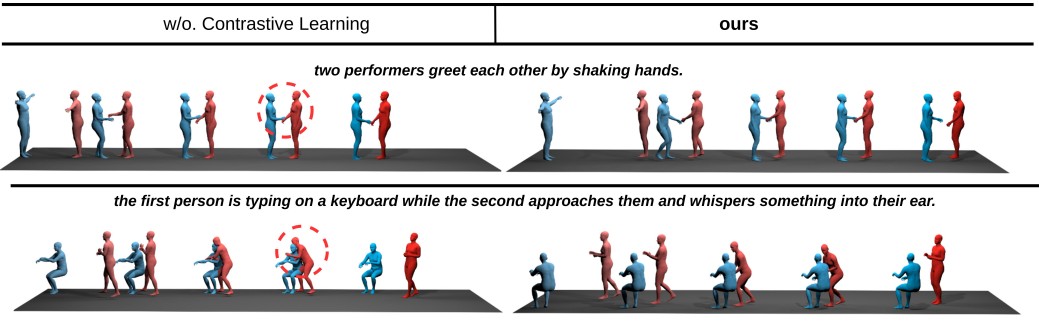

Figure 9: Visualization ablation study for Contrastive learning

## 6.6 Insight into Hierarchical Latents Superiority

We provide both theoretical and empirical evidence that our hierarchical latent design $\mathbf{z} = \{\mathbf{z_o}, \mathbf{z_a}, \mathbf{z_b}\}$ is more effective in the diffusion process than the two-branch baseline $\mathbf{z} = \{\mathbf{z_a}, \mathbf{z_b}\}$.

**Lemma.** For a given diffusion process with denoiser $\theta$, scheduler, and data distribution $p(x)$, higher patch-/channel-wise variance of $x$ leads to more difficult denoising.

**Proof.** The training objective is the expected per-step KL divergence between the true reverse process and the model:

$$\mathcal{L}(\theta) = \mathbb{E}_{p(h)}\mathbb{E}_{p(x_0|h)}\left[\sum_{t=2}^{T} D_{\mathrm{KL}}\big(q(x_{t-1}\mid x_t, x_0)\,\|\,p_\theta(x_{t-1}\mid x_t, h)\big)\right]. \tag{7}$$

Each step has closed-form KL:

$$D_{\mathrm{KL}} = \tfrac{1}{2\sigma_t^2}\,\|\mu_t^\star(x_t, x_0) - \mu_\theta(x_t, t, h)\|_2^2 + \tfrac{1}{2}\left(\tfrac{\tilde{\beta}_t}{\sigma_t^2} - 1 - \log\tfrac{\tilde{\beta}_t}{\sigma_t^2}\right), \tag{8}$$

where schedule-only constants appear in the second term. Hence, channel statistics affect the loss only via the MSE term.

In DDPMs, $\mu_t^\star(x_t, x_0) = A_t x_0 + B_t x_t$. For prediction error $\delta_t = \mu_t^\star - \mu_\theta$,

$$\mathbb{E}_{p(x_0|h)}\|\delta_t\|_2^2 = \mathrm{Tr}(A_t^\top A_t \Sigma_{x|h}) + \|A_t\mu_{x|h} + B_t x_t - \mu_\theta(x_t, t, h)\|_2^2, \tag{9}$$

where $\Sigma_{x|h} = \mathrm{Var}(x_0 \mid h)$. If channel variance $\sigma_h^2$ increases, then $\Sigma_{x|h}$ grows in the PSD sense, making the first term nondecreasing. Thus the expected KL, and consequently $\mathcal{L}(\theta)$, grows with $\sigma_h^2$.

Moreover, since gradient variance scales with $\mathbb{E}\|\delta_t\|_2^2$, larger $\sigma_h^2$ induces noisier gradients, requiring smaller learning rates and slowing convergence. *Therefore, higher channel variance increases both the KL and the optimization difficulty.*

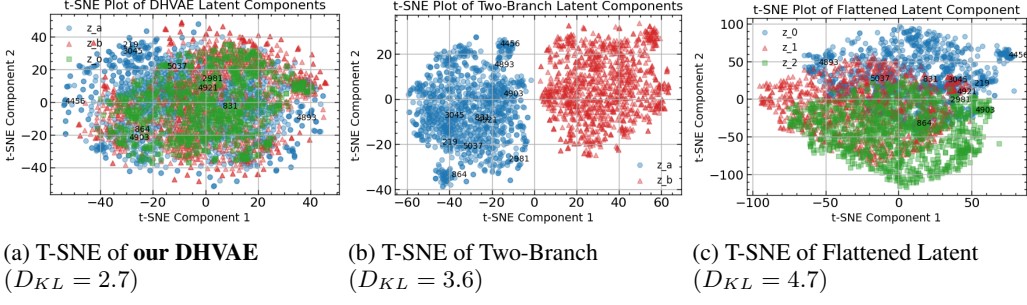

(a) T-SNE of **our DHVAE** ($D_{KL} = 2.7$)

(b) T-SNE of Two-Branch ($D_{KL} = 3.6$)

(c) T-SNE of Flattened Latent ($D_{KL} = 4.7$)

Figure 10: T-SNE projections of latents on the InterHuman test set.

**Fact.** As shown in Fig. 10, our DHVAE produces more uniform covariance across latent channels, while the two-branch and flattened VAEs exhibit noticeable imbalance. Specifically, we compare (a) our hierarchical DHVAE, (b) a two-branch VAE encoding only $\{z_a, z_b\}$, and (c) a flattened VAE following MLD Chen et al. (2023). The improved uniformity stems from our hierarchical decoding scheme, which cascades the shared $z_o$ and global motion sequence (Fig. 2) into the decoders for $z_a$ and $z_b$.

Although the flattened VAE yields lower cross-covariance than the two-branch, its KL divergence and reconstruction quality are substantially worse, leading to poorer generative performance. In contrast, our disentangled hierarchical structure balances variance across components while preserving reconstruction fidelity, resulting in stronger diffusion-based generation. As a fact, with the same penalty $\lambda_{KL} = 0.001$, our DHVAE can encode the latent with the smallest KL Divergence, resulting in more normalized distributions.

Based on the **lemma** above, we further explained why Token Scaling is crucial for denoising, since it can further decrease the covariance magnitude.

### 6.7 ABLATION STUDIES ON CONTACT THRESHOLD

We conduct ablation studies on the contact threshold $\sigma_c$ and the non-contact threshold $\sigma_u$. First, we fix $\sigma_u = 0.30$ and vary $\sigma_c$. When $\sigma_c$ is less than 0.10, the results remain stable for both contact and penetration. However, as $\sigma_c$ increases, the margin between negative and positive samples becomes smaller, making the latent space more tolerant to larger perturbations. This leads to poorer physical plausibility and higher FID scores. Next, we fix $\sigma_c = 0.05$ and vary $\sigma_u$. When $\sigma_u$ is less than 0.2, the model struggles to distinguish between contact and non-contact behaviors. As $\sigma_u$ increases further, the penetration score rises slightly, likely because non-contact motions become more tolerant, which can result in collisions.

| Setting | PV ↓ | PFR ↓ | PDR ↓ | Contact ↑ | FID ↓ |
|---|---|---|---|---|---|
| $\sigma_c = 0.02$, $\sigma_u = 0.30$ | 0.392 | 0.073 | 0.088 | 0.557 | 5.050 |
| $\sigma_c = 0.03$, $\sigma_u = 0.30$ | 0.387 | 0.069 | **0.082** | 0.589 | 5.093 |
| $\sigma_c = 0.05$, $\sigma_u = 0.30$ | 0.390 | **0.064** | 0.087 | 0.581 | **5.015** |
| $\sigma_c = 0.06$, $\sigma_u = 0.30$ | **0.368** | 0.067 | 0.088 | **0.592** | 5.047 |
| $\sigma_c = 0.08$, $\sigma_u = 0.30$ | 0.395 | 0.070 | 0.085 | 0.572 | 5.022 |
| $\sigma_c = 0.10$, $\sigma_u = 0.30$ | 0.402 | 0.068 | 0.090 | 0.524 | 5.035 |
| $\sigma_c = 0.15$, $\sigma_u = 0.30$ | 0.416 | 0.082 | 0.092 | 0.508 | 5.088 |
| $\sigma_c = 0.20$, $\sigma_u = 0.30$ | 0.421 | 0.102 | 0.099 | 0.484 | 5.107 |
| $\sigma_c = 0.05$, $\sigma_u = 0.10$ | 0.417 | 0.109 | 0.092 | 0.584 | 5.044 |
| $\sigma_c = 0.05$, $\sigma_u = 0.20$ | **0.387** | 0.074 | 0.090 | **0.586** | 5.054 |
| $\sigma_c = 0.05$, $\sigma_u = 0.30$ | 0.390 | **0.064** | 0.087 | 0.581 | 5.015 |
| $\sigma_c = 0.05$, $\sigma_u = 0.40$ | 0.395 | 0.066 | **0.085** | 0.577 | **4.997** |
| $\sigma_c = 0.05$, $\sigma_u = 0.50$ | 0.393 | 0.068 | 0.088 | 0.580 | 5.063 |
| $\sigma_c = 0.05$, $\sigma_u = 0.60$ | 0.406 | 0.075 | 0.090 | 0.575 | 5.125 |

Table 7: Performance under different $\sigma_c$ and $\sigma_u$ settings.

### 6.8 PARAMETERS FOR DHVAE

In this section, we evaluate the performance of our method under varying architectural settings to understand its sensitivity to key parameters. Specifically, we analyze the effect of two components: (1) the number of layers in the individual Transformer encoders, and (2) the dimensionality of the latent space.

For the first set of experiments, we fix the latent size to 1 and vary the number of layers in the individual Transformer encoders. For the second, we fix the number of layers to 4 for the individual encoder and 3 for the interaction encoder, while varying the latent size.

As shown in Table 8 and Table 9, we report both the reconstruction performance, measured by reconstruction FID (rFID), and generation quality metrics, including FID, R-Precision, and Multimodal Distance (MM Dist).

We observe that when fixing the latent size, increasing the number of individual encoder layers initially improves the reconstruction quality (i.e., lower rFID), but performance begins to degrade beyond a certain depth, likely due to overfitting or vanishing gradients. Interestingly, the generation performance steadily improves with increasing depth, suggesting that deeper encoders help extract more expressive features for sampling.

The best overall performance is achieved when using 4 layers for the individual encoders and 3 layers for the interaction encoder, as reflected by the lowest FID and highest R-Precision scores. This setting provides a strong trade-off between reconstruction fidelity and generation realism, confirming its use as our default configuration.

| VAE settings | number | rFID ↓ | FID ↓ | R-Prec@1 ↑ | R-Prec@2 ↑ | R-Prec@3 ↑ | MM Dist ↓ |
|---|---|---|---|---|---|---|---|
| | Ground Truth | - | 0.273 | 0.452 | 0.610 | 0.701 | 3.755 |
| | (4, 1) | 0.539 | 5.290 | 0.492 | 0.645 | 0.717 | 3.775 |
| | (4, 3) | 0.503 | **5.015** | 0.496 | 0.647 | 0.720 | 3.772 |
| Layer Number | (4, 5) | 0.522 | 5.145 | 0.489 | 0.643 | 0.712 | 3.780 |
| | (4, 7) | 0.486 | 5.312 | 0.483 | 0.635 | 0.707 | 3.782 |
| | (6, 3) | **0.479** | 5.349 | **0.499** | **0.650** | **0.721** | **3.770** |
| | (8, 3) | 0.507 | 5.612 | 0.483 | 0.637 | 0.715 | 3.784 |

Table 8: Ablation study on different DHVAE layer settings on InterHuman. The tuple in Layer Number represents the number of individual Transformer Encoder layers and the CoTransformer Encoder Layer, respectively.

| VAE settings | number | rFID ↓ | FID ↓ | R-Prec@1 ↑ | R-Prec@2 ↑ | R-Prec@3 ↑ | MM Dist ↓ |
|---|---|---|---|---|---|---|---|
| | l=1 | 0.503 | **5.015** | 0.496 | 0.647 | 0.720 | 3.772 |
| | l=2 | 0.459 | 5.116 | **0.501** | **0.653** | **0.725** | **3.770** |
| Latent Size | l=3 | 0.447 | 5.269 | 0.496 | 0.644 | 0.721 | 3.771 |
| | l=5 | **0.433** | 5.571 | 0.490 | 0.640 | 0.715 | 3.778 |
| | l=7 | 0.462 | 5.670 | 0.486 | 0.638 | 0.709 | 3.782 |

Table 9: Ablation study on different latent sizes for DHVAE on InterHuman.

## 6.9 PARAMETERS FOR DENOISER

In this section, we evaluate the effect of varying the number of layers in the denoiser network. As shown in Table 10, we report the performance changes on both the InterHuman and InterX datasets across several key metrics.

We observe that increasing the depth of the denoiser generally improves R-Precision, while FID tends to stabilize or slightly deteriorate beyond a certain point. To balance generation quality and alignment performance, we empirically choose 13 layers as the optimal configuration for the denoiser in our final model across both datasets.

## 6.10 LIMITATIONS AND DISCUSSION

While our proposed method achieves strong performance across multiple metrics and benchmarks, several limitations remain that highlight important directions for future research.

**Artifacts.** It is worth noting that artifacts still exist in our method, especially on the more challenging InterX dataset, which uses the SMPLX Pavlakos et al. (2019) representation. This representation can more easily lead to foot-sliding, penetrations, or miscontact. Nevertheless, we provide the corresponding visualizations and animations, and we hope our work can inspire further improvements for SMPLX-based datasets.

**Task Scope and Generalization.** Our model is designed for dyadic human-human interactions (i.e., two-person scenarios). As noted in the main text, this formulation inherently restricts its applicability to broader multi-person contexts, such as crowd interactions or team-based tasks. Although

| Dataset | Number of Layers | R-Prec@1 ↑ | R-Prec@2 ↑ | R-Prec@3 ↑ | FID ↓ | MM Dist ↓ | Diversity → |
|---|---|---|---|---|---|---|---|
| | Ground Truth | 0.452 | 0.610 | 0.701 | 0.273 | 3.755 | 7.948 |
| | l=7 | 0.486 | 0.632 | 0.708 | 5.301 | 3.783 | 7.928 |
| | l=9 | 0.491 | 0.642 | 0.718 | 5.107 | 3.772 | 7.970 |
| InterHuman | l=11 | **0.498** | **0.649** | **0.721** | 5.122 | **3.770** | 7.988 |
| | l=13 | 0.496 | 0.647 | 0.720 | 5.015 | 3.772 | **7.952** |
| | l=15 | 0.493 | 0.643 | 0.716 | **4.983** | 3.775 | 7.940 |
| | l=17 | 0.483 | 0.637 | 0.704 | 5.419 | 3.780 | 7.993 |
| | Ground Truth | 0.429 | 0.626 | 0.736 | 0.002 | 3.536 | 9.734 |
| | l=7 | 0.439 | 0.635 | 0.741 | 0.384 | 3.637 | 9.308 |
| | l=9 | 0.443 | 0.637 | 0.742 | 0.366 | 3.602 | 9.157 |
| InterX | l=11 | 0.440 | 0.637 | 0.720 | **0.335** | 3.614 | 9.325 |
| | l=13 | 0.442 | 0.638 | 0.745 | 0.339 | 3.604 | 9.378 |
| | l=15 | **0.445** | **0.642** | **0.748** | 0.378 | **3.600** | 9.265 |
| | l=17 | 0.435 | 0.630 | 0.738 | 0.380 | 3.772 | **9.420** |

Table 10: Performance with different number of Transformer Encoder Layer

this choice reflects the current availability of HHI datasets—most of which contain only paired inter-actions—the development of datasets with dynamic or arbitrary numbers of agents remains a critical bottleneck for extending this line of work. We believe that with richer datasets, the hierarchical design of our method could be adapted to multi-agent scenarios by incorporating scalable inter-agent modeling mechanisms. In fact, one option is to utilize pose estimation methods like Miao et al. (2024) for multi-person scenes and even involve with object Miao et al. (2023) from video.

**Physical Plausibility: Contact.** Although our contrastive learning strategy improves the plausibility of generated interactions by encouraging a well-structured interaction latent space, it does not provide an explicit mechanism to correct fine-grained physical artifacts such as subtle penetrations or missed contacts. These issues are especially prominent in tasks requiring high-precision contact, like handshakes or object passing. Since contrastive learning operates at the latent level and focuses on relational alignment, it lacks direct influence over the decoded physical outcome. We consider integrating post-hoc refinement strategies Wu et al. (2025); Li et al. (2023); Wang et al. (2024) such as Classifier Guidance or contact-aware discriminators (e.g., diffusion-based constraints or physics priors) as promising future directions to further enhance physical realism in a controllable way.

**Evaluation Metrics and Human Judgment.** Our evaluation follows standard protocol using quantitative metrics such as FID, R-Precision, Diversity, and Multimodal Distance. However, these metrics are originally designed for single-agent generation tasks and may fail to capture important aspects of HHI, such as contact quality, synchronization, and plausibility under human judgment. In particular, current metrics are not sensitive to issues like missed hand contact or interpenetration, which often diminish the perceived realism of the motion. We advocate for the development of HHI-specific evaluation protocols—such as plausibility scoring, contact ratio, or perceptual realism assessments—potentially incorporating human-in-the-loop evaluations or learned quality assessors. This will be an integral part of our future work.

**Model Efficiency and Training Cost.** Although our model is lightweight compared to other large-scale Transformer-based generative models, training a variational autoencoder with structured latent diffusion remains computationally demanding, especially when scaling to high-resolution motion or extending to longer sequences. Exploring parameter-efficient variants or leveraging pretraining across different motion domains could alleviate some of these challenges.

**In summary**, while our method demonstrates promising results and novel architectural contributions, we view these limitations as opportunities for further exploration. By addressing the challenges above, future iterations of this framework can evolve into a more versatile and human-aligned HHI generation paradigm.

*both they take out stones, scissors, and cloth while bending over.*

*two performers are practicing fencing, while the first one tries to strike, the second one retreats and guards the counter-attack.*

*the two lift their legs to kick each other simultaneously, followed by the first one attempting another assault.*

*they make a respectful greeting by nodding their heads towards each other.*

*one person moves forward and strikes the other person with a right hook, and the other person retaliates by striking the other person with their right leg.*

*one person spreads both arms and prepares to do a backbend, and the other person approaches and pushes against one person's chest.*

Figure 11: HHI motion sequences generated on InterHuman dataset

## 6.11 MORE VISUALIZATION FOR BOTH INTERHUMAN & INTERX

Here we visualize more results for both InterHuman and InterX in Fig. 11 and Fig. 12, demonstrating the comprehensive high-quality of the HHI sequence. For InterX visualization, not only the main body parts, our DHVAE can generate dedicated **gestures** as well, suggesting the full body HHI generation ability and generalizability.

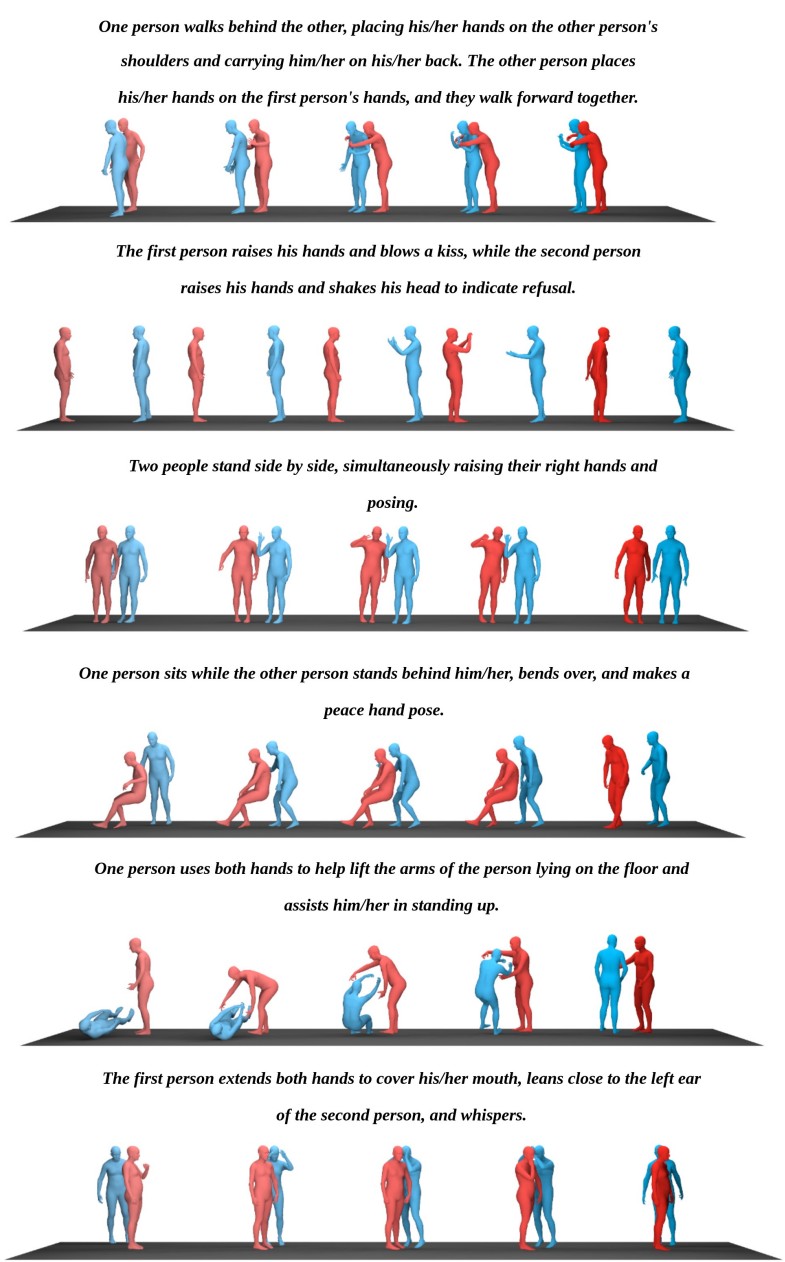

Figure 12: HHI motion sequences generated on InterX dataset, including gestures

## 6.12 METRICS

**FID**: Frechet Inception Distance is used to evaluate the dissimilarity between two distributions as

$$d_F(\mathcal{N}(\mu, \boldsymbol{\Sigma}), \mathcal{N}(\mu', \boldsymbol{\Sigma}'))^2 =$$
$$||\mu - \mu'||^2 + Tr(\boldsymbol{\Sigma} + \boldsymbol{\Sigma}' - 2\sqrt{\boldsymbol{\Sigma}\boldsymbol{\Sigma}'}), \tag{10}$$

where the $\mu$ and $\mu'$ are the feature's mean values of generated samples and ground truth, and $\boldsymbol{\Sigma}$ and $\boldsymbol{\Sigma}'$ are the feature's covariance matrices respectively. This metric measures the distance between two normal distributions, that is generated samples and the ground truth. The less the FID is, the better the model performs. Since generated motion results usually contain hundreds or even thousands of frames, FID for motion samples should be calculated with extracted features instead of raw generated motion data and ground truth. In practice, pre-trained action recognition models by Guo et al. Guo et al. (2020) and by Ji et al. Ji et al. (2018) are utilized to extract features from input samples.

**Diversity**: Diversity measures the variance of the generated motions across all action categories. From a set of all generated motions from various action types, two subsets of the same size $S_d$ are randomly sampled, and their respective feature sets are extracted as $\{\mathbf{f_1}, \cdots, \mathbf{f_{S_d}}\}$ and $\{\mathbf{f_1'}, \cdots, \mathbf{f_{S_d}'}\}$. The diversity between them is given by:

$$\text{Div} = \frac{1}{S_d} \sum_{i=1}^{S_d} ||\mathbf{f_i} - \mathbf{f_i'}||^2 \tag{11}$$

Usually, good generated samples are supposed to have a similar diversity value as ground truth.

**Multimodality**: Different from diversity, multimodality measures how much the generated motions diversify within each action type. Given a set of motions with $C$ action types. For $c$-th action, we randomly sample two subsets with same size $S_l$, and then extract two subset of feature vectors $\{\mathbf{f}_{c,1}, ..., \mathbf{f}_{c,S_l}\}$ and $\{\mathbf{f}_{c,1}', ..., \mathbf{f}_{c,S_l}'\}$. The multimodality of this motion set is formalized as

$$\text{Multimodality} = \frac{1}{C \cdot S_l} \sum_{c=1}^{C} \sum_{i=1}^{S_l} ||\mathbf{f_{c,i}} - \mathbf{f_{c,i}'}||^2 \tag{12}$$

**Multimodal Distance**: This metric describes how close the relationship is between the motion features and the text features. It is computed as the average Euclidean distance between the generated motion features and corresponding text features. For given text $\mathbf{T}$ with feature $\mathbf{f_T}$ and corresponding generated motion $\mathbf{M}$ with feature $\mathbf{f_M}$. The multimodal distance is given by:

$$\text{MMD} = \sqrt{\frac{1}{n} \sum_{i=1}^{n} ||\mathbf{f_{T,i}} - \mathbf{f_{M,i}}||^2} \tag{13}$$

where $n$ is the sample number of text $\mathbf{T}$. In practice, the text features are extracted by a pre-trained text encoder by Guo et al. Guo et al. (2022a). Multimodal distance is also applied for motion generation tasks driven by other modalities than text, such as music and audio.

**R-Precision**: Also known as motion-retrieval precision, it calculates the text and motion's top $K$ matching accuracy among $R$ documents based on the Euclidean distance. Actually, most works follow the method of Guo et al. Guo et al. (2022a). For each generated motion, its ground-truth text description and 31 randomly selected mismatched descriptions from the test set form a description pool. This is followed by calculating and ranking the Euclidean distances between the motion feature and the text feature of each description in the pool. Then accuracy will be calculated for the top 1, 2, and 3 places.

## 6.13 DETAILED PROOF OF OBJECTIVE FUNCTION

While Equation 1 does model the joint probability $p(\mathbf{x}_a, \mathbf{x}_b)$ in a standard VAE framework, we note that in practice, conventional implementations often assume conditional independence given the latent variable $\mathbf{z}$, which represents their global semantic and interaction features, i.e.,

$$p(\mathbf{x}_a, \mathbf{x}_b | \mathbf{z}) = p(\mathbf{x}_a | \mathbf{z}) \, p(\mathbf{x}_b | \mathbf{z}).$$

Under this factorization, the model cannot capture the direct dependencies between agents' motions. Therefore, the statement in the paper that "Equation 1 ignores the conditional dependency between the agents" refers to this practical limitation rather than the mathematical form of the joint distribution. Our DHVAE explicitly models such dependencies through its CoTransformer structure. Here we assume that under a given $z_o$, $x_a$ and $x_b$ are conditionally independent. The full proof with our assumption is: Start from the marginal log-likelihood:

$$\log p(\mathbf{x}_a, \mathbf{x}_b) = \log \int p(\mathbf{x}_a, \mathbf{x}_b, \mathbf{z}_a, \mathbf{z}_b, \mathbf{z}_o)\, d\mathbf{z}_a\, d\mathbf{z}_b\, d\mathbf{z}_o.$$

Introduce the variational posterior:

$$q(\mathbf{z}_a, \mathbf{z}_b, \mathbf{z}_o \mid \mathbf{x}) = q(\mathbf{z}_a \mid \mathbf{x}_a)\, q(\mathbf{z}_b \mid \mathbf{x}_b)\, q(\mathbf{z}_o \mid \mathbf{z}_a, \mathbf{z}_b).$$

By Jensen's inequality:

$$\log p(\mathbf{x}_a, \mathbf{x}_b) \geq \mathbb{E}_q\Big[ \log p(\mathbf{x}_a, \mathbf{x}_b, \mathbf{z}_a, \mathbf{z}_b, \mathbf{z}_o) - \log q(\mathbf{z}_a, \mathbf{z}_b, \mathbf{z}_o \mid \mathbf{x}) \Big].$$

We assume a conditional independence structure in the generative process, where the individual motions $x_a$ and $x_b$ are conditionally independent given their respective local latents and the shared interaction latent. Under this assumption, the joint distribution factorizes as:

$$p(\mathbf{x}_a, \mathbf{x}_b, \mathbf{z}_a, \mathbf{z}_b, \mathbf{z}_o) = p(\mathbf{z}_a)\, p(\mathbf{z}_b)\, p(\mathbf{z}_o)\, p(\mathbf{x}_a \mid \mathbf{z}_o, \mathbf{z}_a)\, p(\mathbf{x}_b \mid \mathbf{z}_o, \mathbf{z}_b),$$

and rearranging terms yields:

$$\begin{aligned}
\log p(\mathbf{x}_a, \mathbf{x}_b) \geq\ & \mathbb{E}_q\big[ \log p(\mathbf{x}_a \mid \mathbf{z}_o, \mathbf{z}_a) + \log p(\mathbf{x}_b \mid \mathbf{z}_o, \mathbf{z}_b) \big] \\
& - D_{\mathrm{KL}}\big(q(\mathbf{z}_a \mid \mathbf{x}_a)\|p(\mathbf{z}_a)\big) - D_{\mathrm{KL}}\big(q(\mathbf{z}_b \mid \mathbf{x}_b)\|p(\mathbf{z}_b)\big) \\
& - D_{\mathrm{KL}}\big(q(\mathbf{z}_o \mid \mathbf{z}_a, \mathbf{z}_b)\|p(\mathbf{z}_o)\big).
\end{aligned}$$

## 6.14 DETAILED DIAGRAM FOR THE DHVAE

Here we show a more detailed version of our DHVAE with both the CoTransformer and individual encoders/decoders.

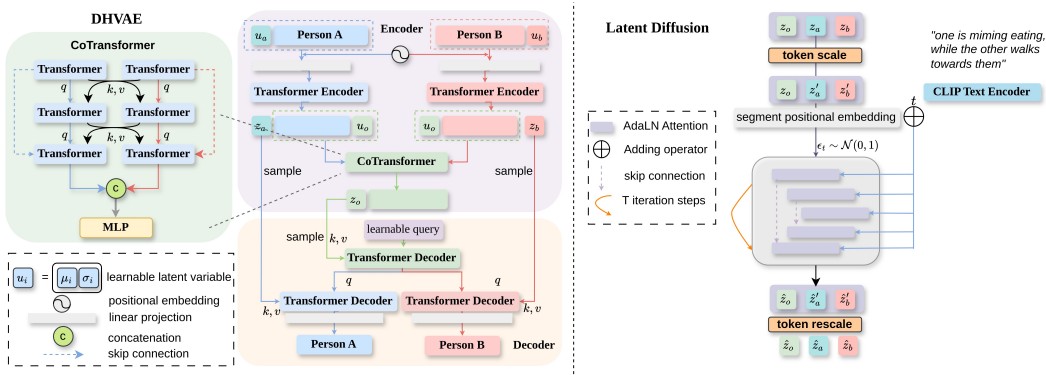

Figure 13: Detailed diagram of DHVAE

## 6.15 USE OF LLMS

In this work, we made limited use of large language models (LLMs), specifically GPT-5 and Gemini, only for paper polishing and grammar refinement. There are no parts of the research design, core ideas, experimental implementation, analysis, or conclusions that were generated by the LLMs. All technical contributions, model development, experiments, and interpretations were edited, executed, and validated exclusively by our human authors. The use of LLMs was restricted to improving readability and presentation quality, without influencing the scientific content or originality of the work.

