# OpenReview forum: "Disentangled Hierarchical VAE for 3D Human-Human Interaction Generation"
_ICLR.cc/2026/Conference — ICLR 2026 Poster_

### Official Review · Reviewer_7SSr · 2025-10-15

**Soundness:** 3
**Presentation:** 3
**Contribution:** 3
**Rating:** 8
**Confidence:** 5

**Summary:**

The paper presents a novel approach to generate two person interaction motion from text called DHVAE. DHVAE obtain encoding for both motion then fuse these encoding to obtain an interaction encoding. Then a positive/negative pair of encoding is created to help learning the interaction trought a triplet loss. Finally, diffusion is used to denoise the interaction encoding that can then bit split into two motion encoding for both humans. Extensive experiments show that the method beat SOTA on two commonly used datasets quantitatively and qualitatively.

**Strengths:**

- In the context of the datasets used creating positive and negative encoding is a good idea to limit body penetration.
- method and implementation are clear and well explained
- extensive experiment and ablation show the superiority of the method
- Qualitative results are convincing.

**Weaknesses:**

- It is not clear how "contact" is determined. Is it siply trought distance between the different body part ?
- While the idea of building positive and negative encoding is nice it is limited to direct physical interaction or simple non physical interaction (which comprise nearly all current interaction datasets). But at the same time all lot of human interaction are indirect  (e.g a person point somewhere and the other turn to look) and a lot more subtle. How would this method works in those contexts ? Especially how will the negative encoding work for non contact action when 45–90cm translation might not "break the interaction ?
- In table 1 we see that the proposed method has the lowest multimodality on interhuman while the results are not as bad on interx but there is no discussion of this in the paper. How do these results affect the generation qualitatively ?
- In the supplementary video we can see that penetration still happens in the sample generated by the proposed method. How could this be mitigated using the positive/negative encoding. An ablation on different value for the various  "σ's" used  and the on how contact are decides could have been interesting to see how much the penetration issue can been mitigated.
- I might be wrong but in in the appendix the authors consider InterLDM [1] "to be the first application of latent diffusion models to the HHI task" but [2] appear to be older.

[1]Boyuan Li, Xihua Wang, Ruihua Song, and Wenbing Huang. Two-in-one: Unified multi-person interactive motion generation by latent diffusion transformer. In ICASSP 2025-2025 IEEE International Conference on Acoustics, Speech and Signal Processing (ICASSP)
[2] CHOPIN, Baptiste, TANG, Hao, et DAOUDI, Mohamed. Bipartite graph diffusion model for human interaction generation. In : Proceedings of the IEEE/CVF Winter Conference on Applications of Computer Vision. 2024

**Questions:**

See Weaknesses.

---

> ### Author Response · Authors · 2025-11-23
> **Weakness 1, 2**
>
> We sincerely thank Reviewer 7SSr for the encouraging and detailed feedback. We greatly appreciate your positive evaluation of our method, presentation, and experiments, as well as your insightful comments on its limitations and potential extensions. We have carefully considered each of your suggestions and provide detailed responses below.
>
>
>
> > Weakness 1: How "contact" is determined?
>
> Thank you for the valuable question. In our implementation, contact is not defined merely by joint-to-joint distances, as human interactions often occur across arbitrary surface regions rather than specific keypoints. Instead, we compute contact through voxel-level overlap between the two participants after voxelizing their full-body meshes. Any non-zero intersection between the voxelized bodies is treated as contact.
>
> We also note that certain interaction patterns—such as slashing or near-miss actions—may involve very close proximity without actual surface intersection. To robustly capture such “almost contact” cases while avoiding missed detections, we apply a two-voxel dilation to the voxelized bodies before overlap testing. This ensures that near-contact interactions are correctly recognized without relying on expensive inverse-kinematics–based surface queries.
>
> Finally, because frame-by-frame IK fitting is computationally heavy, especially on the InterHuman Dataset, therefore, we annotated them with contact label in advance to ensure training efficiency. We have added additional discussion to clarify this in the latest manuscript Line 203-204.
>
>
>
>
>
> >  Weakness 2: How will the negative encoding work for non contact action when 45–90cm translation might not "break the interaction ?
>
> We appreciate this insightful question. Indirect and subtle interactions—such as pointing, gaze following, or small gestures—are indeed different from direct physical contact. In our framework, however, these interactions are largely captured by the global and individual latent representations, which encode higher-level semantic cues through the hierarchical structure rather than relying solely on contact patterns. Such subtle signals are typically less sensitive to small spatial perturbations compared with physically grounded interactions like penetration or mis-contact.
>
> For constructing positive samples in indirect interactions (e.g., pointing), applying a small shift (0–30 cm) does not substantially alter the semantic intent and therefore remains consistent with the original interaction. For negative samples, although a 45–90 cm displacement may not completely “break” certain non-contact interactions from a human perspective, it provides a sufficiently strong perturbation to expand the representation boundary and enforce robustness. This large-shift augmentation helps the model learn to distinguish meaningful interaction cues even when explicit contact is absent. Table 5 of the latest revision
> suggests that when the $\sigma_c$ between 0.2-0.4 is a reasonable setting for both physical plausibility and FID.
>
>
>
> ### Table 7. Performance under different σc and σu settings
> *Suggested reasonable range: 0.02 < σc < 0.10*
>
> | Setting                       | PV ↓   | PFR ↓ | PDR ↓ | Contact ↑ | FID ↓  |
> |-------------------------------|--------|-------|-------|-----------|--------|
> | σc = 0.02, σu = 0.30          | 0.392  | 0.073 | 0.088 | 0.557     | 5.050  |
> | σc = 0.03, σu = 0.30          | 0.387  | 0.069 | **0.082** | 0.589     | 5.093  |
> | σc = 0.05, σu = 0.30          | 0.390  | **0.064** | 0.087 | 0.581     | **5.015** |
> | σc = 0.06, σu = 0.30          | **0.368** | 0.067 | 0.088 | **0.592** | 5.047  |
> | σc = 0.08, σu = 0.30          | 0.395  | 0.070 | 0.085 | 0.572     | 5.022  |
> | σc = 0.10, σu = 0.30          | 0.402  | 0.068 | 0.090 | 0.524     | 5.035  |
> | σc = 0.15, σu = 0.30          | 0.416  | 0.082 | 0.092 | 0.508     | 5.088  |
> | σc = 0.20, σu = 0.30          | 0.421  | 0.102 | 0.099 | 0.484     | 5.107  |
> | **σc = 0.05 sweep (σu variable)** |       |       |       |           |        |
> | σc = 0.05, σu = 0.10          | 0.417  | 0.109 | 0.092 | 0.584     | 5.044  |
> | σc = 0.05, σu = 0.20          | **0.387** | 0.074 | 0.090 | **0.586** | 5.054  |
> | σc = 0.05, σu = 0.30          | 0.390  | **0.064** | 0.087 | 0.581     | 5.015  |
> | σc = 0.05, σu = 0.40          | 0.395  | 0.066 | **0.085** | 0.577     | **4.997** |
> | σc = 0.05, σu = 0.50          | 0.393  | 0.068 | 0.088 | 0.580     | 5.063  |
> | σc = 0.05, σu = 0.60          | 0.406  | 0.075 | 0.090 | 0.575     | 5.125  |

---

> ### Author Response · Authors · 2025-11-23
> **Weakness 3, 4, 5**
>
> > Weakness 3: Discussion on multimodality difference between InterHuman and InterX
>
> Thanks for the insightful observation regarding the different multimodality behaviors between InterHuman and InterX. Indeed, most models have higher multimodality on the InterX dataset than InterHuman, and this largely stems from intrinsic differences between the datasets. Specifically, the InterHuman dataset constrains one of the actors to initially face the Y-axis at the origin, which implicitly reduces variation in the motion space. In contrast, InterX imposes no constraint on the initial facing direction, resulting in significantly greater diversity.
>
> As a result, even though our method attempts to overfit the training set for both datasets, the multimodality measured on InterHuman remains relatively low because the initial conditions restrict the diversity of possible generations. For InterX, however, the unconstrained initial facing directions produce inherently diverse motion patterns, so features across different generations can vary substantially as animated in our visualization of InterX, leading to higher multimodality scores. Qualitatively, this difference mainly manifests in the variation of the facing direction in the generated motions.
>
>
> > Weakness 4: Ablations on different sigma
>
> Thank you so much for your professional suggestions concerning the contrastive learning.
> We conduct ablation studies on the contact radius $\sigma_c$ and the non-contact radius $\sigma_u$. First, we fix $\sigma_u=0.30$ and vary $\sigma_c$. When $\sigma_c$ is less than 0.10, the results remain stable for both contact and penetration. However, as $\sigma_c$ increases, the margin between negative and positive samples becomes smaller, making the latent space more tolerant to larger perturbations. This leads to poorer physical plausibility and higher FID scores. Next, we fix $\sigma_c=0.05$ and vary $\sigma_u$. When $\sigma_u$ is less than 0.2, the model struggles to distinguish between contact and non-contact behaviors. As $\sigma_u$ increases further, the penetration score rises slightly, likely because non-contact motions become more tolerant, which can result in collisions.
>
>
> ### Table 7. Performance under different σc and σu settings
> *Suggested reasonable range: 0.02 < σc < 0.10*
>
> | Setting                       | PV ↓   | PFR ↓ | PDR ↓ | Contact ↑ | FID ↓  |
> |-------------------------------|--------|-------|-------|-----------|--------|
> | σc = 0.02, σu = 0.30          | 0.392  | 0.073 | 0.088 | 0.557     | 5.050  |
> | σc = 0.03, σu = 0.30          | 0.387  | 0.069 | **0.082** | 0.589     | 5.093  |
> | σc = 0.05, σu = 0.30          | 0.390  | **0.064** | 0.087 | 0.581     | **5.015** |
> | σc = 0.06, σu = 0.30          | **0.368** | 0.067 | 0.088 | **0.592** | 5.047  |
> | σc = 0.08, σu = 0.30          | 0.395  | 0.070 | 0.085 | 0.572     | 5.022  |
> | σc = 0.10, σu = 0.30          | 0.402  | 0.068 | 0.090 | 0.524     | 5.035  |
> | σc = 0.15, σu = 0.30          | 0.416  | 0.082 | 0.092 | 0.508     | 5.088  |
> | σc = 0.20, σu = 0.30          | 0.421  | 0.102 | 0.099 | 0.484     | 5.107  |
> | **σc = 0.05 sweep (σu variable)** |       |       |       |           |        |
> | σc = 0.05, σu = 0.10          | 0.417  | 0.109 | 0.092 | 0.584     | 5.044  |
> | σc = 0.05, σu = 0.20          | **0.387** | 0.074 | 0.090 | **0.586** | 5.054  |
> | σc = 0.05, σu = 0.30          | 0.390  | **0.064** | 0.087 | 0.581     | 5.015  |
> | σc = 0.05, σu = 0.40          | 0.395  | 0.066 | **0.085** | 0.577     | **4.997** |
> | σc = 0.05, σu = 0.50          | 0.393  | 0.068 | 0.088 | 0.580     | 5.063  |
> | σc = 0.05, σu = 0.60          | 0.406  | 0.075 | 0.090 | 0.575     | 5.125  |
>
> > Weakness 5: InterLDM vs. BiGraphDiff
>
> Thank you for your attention to related work and for your careful reading. In our opinion, BiGraphDiff has overlap but cannot be considered as a latent diffusion model because (1) its operations remain in the graph space, which essentially represents raw skeletal data rather than a learned latent representation; (2) it does not employ self-supervised or unsupervised learning to compress motion into a compact latent space.
>
> In contrast, the InterLDM leverages a VAE-based self-supervised compression, which significantly improves inference efficiency and aligns with the general definition of latent diffusion models. We also welcome your discussion on this!

---

### Official Review · Reviewer_p1j7 · 2025-10-28

**Soundness:** 3
**Presentation:** 3
**Contribution:** 2
**Rating:** 4
**Confidence:** 3

**Summary:**

This paper proposes encoding the motions of two people into an interaction-aware variable $z_{o}$ in the latent space, and employs a contrastive learning strategy to improve the quality of interactions during reconstruction.

**Strengths:**

1. The use of contrastive learning for two-person interaction generation is theoretically sound and has been validated in other motion generation domains (e.g., sign-language synthesis).
2. The paper’s core contributions, the global interaction latent variable $z_{o}$ and the CoTransformer are both empirically verified through experiments.
3. The manuscript is clearly written; the authors’ intentions can be readily followed from the text, and the Contrastive Learning algorithm is presented in a transparent and unambiguous manner.

**Weaknesses:**

1. The core contribution of the paper is rather narrow: encoding two-person motions into a single latent variable $z_{o}$ improves generation quality, but it does not yield creative or novel motion combinations.
2. The effectiveness of the contrastive-loss term is not reflected in the numerical metrics; the authors attempt to demonstrate its role through PCA visualizations in Appendix 6.5 (Figs. 7 & 8) and qualitative renderings. **Nevertheless, further evidence is required, as the high-quality results may primarily stem from the KL-divergence regularization.**
3. Although the method enhances human-human interaction generation, **it offers little insight into scaling motion synthesis to more than two agents**. Moreover, the adopted contrastive-learning strategy poses new challenges for defining multi-person interactions: **how should “contact” and “away” be characterized among multiple individuals engaged in sustained interaction?**

**Questions:**

1. The Contrastive Learning itself may destroy the original distance information between the two actors; if in a sequence their inter-distance changes drastically (from far to close or vice-versa), will the contrastive loss still work satisfactorily?
2. The authors provide abundant visual results to demonstrate the effectiveness of Contrastive Learning, yet **an additional user study would make the claim more convincing**.
3. In view of the concerns about the scalability of Contrastive Learning and its actual contribution to quality, I tentatively assign a rating of 4. **If the authors supply strong evidence that contrastive learning brings substantial quality gains, can be extended to multi-person scenarios, or notably improves cases with large distance variations, I will raise my rating**.

---

> ### Author Response · Authors · 2025-11-23
> **Weakness 1, 2**
>
> We sincerely thank Reviewer p1j7 for their comments and useful suggestions, including the suggestion of scalability to multi-person cases.
>
> > Weakness 1: The core contribution of the paper is rather narrow: encoding two-person motions into a single latent variable $z_o$ improves generation quality, but it does not yield creative or novel motion combinations.
>
> Our core contributions are two-fold: we first propose the *disentangled hierarchical* encoding–decoding scheme, and first *embed contrastive learning* that enhances physical plausibility for HHI.
>
> The single latent $z_o$ is actually the global interaction latent and does not explicitly encode two-person motions into a single latent. It encodes the interaction between the two agents. The motion latent $z_a$ (of one agent) and motion latent $z_b$ (of the other agent) remain disentangled. That is, we encode the individual actions separately, hence our method is referred to as *disentangled*. We will further clarify this.
>
> Again, we thank you for raising the challenging problem of generating novel motion combinations. From our perspective, the creative samples mean the out-of-bag or with unseen prompts.
> Generating Out-of-bag / novel samples is always a main challenge in generative models, and no external knowledge is involved, especially when the dataset size is small. In the Paper [1], the authors conclude that "an exponential need for training data which implies that the key to “zero-shot” generalization capabilities under large-scale training data and compute paradigms remains to be found."
>
> Compared with image or audio generation whose training set with augmentation could take up to ten millions, InterHuman and InterX is only about 10 \textbf{thousand}, which makes generating novel motion combniation even harder. Compared with previous work, InterMask and TIMotion, our \textbf{results over the test dataset with unseen prompts} outperform them marginally, indicating high-fidelity (FID) and text-aligned (R-precisions) Out-of-bag generation ability and generalizability.
>
> Also note that our design choices are grounded in real gaps in the literature. Our hierarchical encoding significantly improves semantic consistency, and our contrastive learning effectively alleviates interpenetration issues. We believe that such effective and reproducible innovations are very beneficial for advancing research in this direction.
>
> [1] Udandarao, Vishaal, et al. "No" zero-shot" without exponential data: Pretraining concept frequency determines multimodal model performance." Advances in Neural Information Processing Systems 37 (2024): 61735-61792.
>
>
>
> > Weakness 2: The effectiveness of the contrastive-loss term is not reflected in the numerical metrics; the authors attempt to demonstrate its role through PCA visualizations in Appendix 6.5 (Figs. 7 \& 8) and qualitative renderings. Nevertheless, further evidence is required, as the high-quality results may primarily stem from the KL-divergence regularization.
>
>
> The contrastive learning is mainly designed to improve physical plausibility, e.g., the penetration and contact, as evidenced in
> Table 5 in the supplementary (Table 4 in Section 4.1 of the latest revision), which significantly reduces penetration and increases contact ratio.
>
>
> Table 4. Penetration metrics and contact ratio for state-of-the-art models.
>
> | Model            | PV ↓   | PFR ↓  | PDR ↓  | Contact ↑ |
> |-----------------|--------|--------|--------|-----------|
> | MLD*             | 0.503  | 0.108  | 0.175  | 0.427     |
> | InterMask        | 0.873  | 0.149  | 0.243  | 0.349     |
> | TIMotion         | 0.485  | 0.122  | 0.104  | 0.466     |
> | Ours w/o triplet | 0.446  | 0.107  | 0.102  | 0.445     |
> | **Ours**         | **0.390**  | **0.064**  | **0.087**  | **0.581**     |
>
>
> Compared with MLD and InterLDM, which use only KL-divergence as constraints, our method still achieves marginally better performance in terms of FID, Diversity (fidelity), MM-Distance, and R-presions (semantic alignment), indicating that the improvement is not solely due to the KL-divergence but also benefits from our hierarchical structure. Moreover, since our performance on these metrics is approaching the upper bound defined by the training dataset, the additional gain from contrastive learning is limited in terms of standard metrics.

---

> ### Author Response · Authors · 2025-11-23
> **Weakness 3 - Question 3**
>
> > Weakness 3 & Question 3: Multi-Person Interaction.
>
> Multi-person interaction is indeed a more generalized approach, but for the future. Current research is focusing on Two-Person Interaction, including InterMask (ICLR 2025), TIMotion (CVPR 2025). Another problem is that current benchmarks are designed for Two-Person generation, and as far as we know, there is no benchmark for Multi-Person interaction. Our current method is designed and evaluated specifically for dual-agent interaction. In summary, due to the limitations of the benchmark and consistency with prior work, we do not claim direct applicability to three-agent interaction.
>
> Nevertheless, we believe our method can be scaled to multi-person scenarios. For example, in a 3-person case ${a, b, c}$, there would be seven latent variables: one overall latent $z_o$ for all three participants, three paired latents $(z_{a,b}, z_{a,c}, z_{b,c})$, and three individual latents $(z_{a}, z_{b}, z_{c})$. The encoder would process these hierarchically, and the decoder would reconstruct each individual conditioned on their corresponding latent and relevant paired latents, e.g.,
> $$
> q(x_a \mid z_a, z_{a,b}, z_{a,c}, z_o)
> $$
>
> Contrastive learning would be applied to the paired latents $z_{a,b}, z_{a,c}, z_{b,c}$.
>
>
> This strategy can also extend to four or more participants, just with more paired latents. The encoder branch will combine corresponding paired latents as individual latents, while the decoder will decode with::
>
> $$q(x_a \mid z_a, z_{a,b}, z_{a,c}, z_{a,d}, z_o).$$
>
> In cases where a participant is absent, a masked input can be used to indicate the missing individual. As far as we know, previous works such as InterMask and TIMotion which is particularly desinged for two-person interaction generations, have no such flexibility to scale their model to 3 or more people cases.
>
> However, due to the lack of a Multi-person dataset and benchmark, it remains a future task.
>
>
>
> > Question 1: The Contrastive Learning itself may destroy the original distance information between the two actors; if in a sequence their inter-distance changes drastically (from far to close or vice-versa), will the contrastive loss still work satisfactorily?
>
> Thanks for your comments on our contrastive learning strategies. We agree that the dramatic distance change will make the training more difficult, but our negative sample mitigates such cases by learning a margin between implausible contact, especially penetrations. As shown in Table 5 of the supplementary, compared with MLD, InterMask, and TIMotion, and our own variant without contrastive learning, our DHVAE with contrastive learning achieves the lowest penetration score, indicating better physical plausibility. Besides, in the visualization of shaking hands and whispering (Fig. 3, groups 1 and 3), our agents walk towards each other from a long distance (about 3 meters), suggesting that our methods are capable of generating such close interaction motions with dramatic changes, which looks better than TIMotion and InterMask. We will highlight this in the text.
>
>
> > Question 2: The authors provide abundant visual results to demonstrate the effectiveness of Contrastive Learning, yet an additional user study would make the claim more convincing?
>
> That's a great idea. We also believe that a user study would substantially strengthen our work and make the effectiveness of our contrastive learning more convincing. Therefore, we plan to conduct a dedicated user study with public voting to evaluate the contrastive learning method. Due to confidentiality considerations, we will carry this out after the rebuttal period, and we are looking forward to your participation!

---

> > ### Comment · Reviewer_p1j7 · 2025-11-26
> >
> > I appreciate the authors' efforts in addressing my questions. I acknowledge that the paper contributes to improving the physical realism of HHI interactions. However, my concern regarding HHI is more about whether it can scale to multiple people and diverse actions within the limited data scope. The authors also acknowledge that their work does not claim direct applicability to three-agent interaction. Moreover, their results on the test dataset with unseen prompts only marginally outperform other works. The absence of a user study at this stage also fails to demonstrate human evaluation of the motion results. Therefore, unfortunately, while the authors have demonstrated that the current work can improve motion quality to some extent, this work is "not interesting enough" and contributes insufficiently to future research expansion. I will therefore maintain my current score and increase my confidence. Collecting two-person motion data is already extremely difficult; must we wait until we have sufficient multi-person motion data before conducting related research?

---

> > > ### Author Response · Authors · 2025-11-26
> > >
> > > We appreciate your acknowledgment of our contributions toward improving **physical plausibility** (penetration by about *20%*, and contact by about *25%* relatively on InterHuman dataset) and **semantic alignment** (especially on InterX dataset, with R-precision-3 *7.3%*, and FID by *11.2%* relatively ).
> > >
> > > Regarding  “results on the test dataset with unseen prompts only marginally outperform other works,” is not entirely correct. Compared to the nearest competitor TIMotion (CVPR 2025), as shown in *Table 1*, we improve the R-Prec@1 by *7.3%* (0.412 -> 0.442) on IterX dataset and by 2.3% (0.485 -> 0.496) on InterHuman dataset. And we improve the FID by *11.9%* (0.385 -> 0.339) on InterX dataset and by 2.7% (5.153 -> 5.015 for InterMask) on InterHuman dataset.
> > >
> > > Regarding the reviewer's comment "must we wait until we have sufficient multi-person motion data before conducting related research?" Yes, we believe at least a multi-person evaluation benchmark should be available. Developing a new benchmark for multi-person interaction is an interesting direction but out of the scope of this paper. Clearly there are still research gaps that must be filled in two-person interaction and this paper makes significant advancements in that direction. Our paper addresses major research gaps such as **physical plausibility** and **semantic alignment**, proposes a novel method and reports better results than existing SOTA.
> > >
> > > We emphasize that two-person interaction is still an active research area and recent works like TIMotion (CVPR 2025), InterMask (ICLR 2025), in2IN (CVPRW 2025) are all conducted on these two-person interaction datasets.
> > >
> > > Finally, we thank you for suggesting the user study experiment. We are currently still conducting user study and will report the results as soon as they are available.

---

### Official Review · Reviewer_h5hK · 2025-10-28

**Soundness:** 2
**Presentation:** 2
**Contribution:** 2
**Rating:** 4
**Confidence:** 4

**Summary:**

This paper proposes an improved VAE structure, namely DHVAE, for better Human-Human Interaction (HHI) synthesis. Specifically, the individual motions in HHI are separately encoded into latent space and then fused, resulting in two latent variables representing individual motions and one latent variable $z_o$ representing global information. These latent variables are used to train a denoiser that learns to generate HHI-representative latent variables from text. Additionally, the paper proposes Interaction Contrastive Learning to train $z_o$ with the goal of achieving physically plausible results. The denoiser design also incorporates corresponding techniques to adapt to the proposed VAE structure. Quantitative results demonstrate advantages over many existing methods.

**Strengths:**

1. The figures and tables are well-presented, and the appendix is relatively comprehensive.

2. The method shows advantages in quantitative metrics on both InterHuman and InterX datasets.

**Weaknesses:**

**Major**

1. **The contribution of the main component—DHVAE—has not been sufficiently discussed.** Compared to "a flat, unified latent representation" VAE, DHVAE uses three latent variables to represent an HHI sequence. Does this mean that the dimensionality of DHVAE's latent variables is three times that of a standard VAE? For example, on the InterHuman dataset, is D(DHVAE) : D(VAE) = 3×256 : 1×256? If so, are the comparisons in Table 2 and Table 3 fair?

2. **The paper's presentation and formulations lack rigor.** For example, at Line 178, the paper states that Equation 1 "ignores the conditional dependency between the agents." However, Equation 1 actually models the joint probability between $x_a$ and $x_b$.

   Furthermore, **Equation regarding the ELBO of $\log p(x_a, x_b)$ lacks rigorous derivation.** Without a formal proof showing that this objective is indeed a valid lower bound of the log-likelihood, it should be treated as a heuristic training objective rather than a theoretically grounded ELBO. The authors should either provide the complete derivation or clarify the assumptions under which this bound holds.

3. **Insufficient evidence for the contribution of Interaction Contrastive Learning (ICL).** Table 3 shows that ICL does not significantly help with FID and R-Prec metrics; Table 5 demonstrates that ICL can reduce penetration. However, a critical experiment is missing: proving that ICL does not reduce penetration simply by pushing people further apart—it should be demonstrated that the introduction of ICL at least does not decrease the contact ratio, i.e., maintaining the contact ratio while reducing penetration.

4. **Despite achieving certain quantitative results, the generated effects still exhibit many artifacts based on the provided supplementary materials,** such as penetration, foot sliding, poor contact, and other physically implausible behaviors. This undoubtedly undermines the paper's claim that "DHVAE achieves superior physical plausibility."

**Minor**

5. **The proposed Interaction Contrastive Learning does not demonstrate its contribution in the main experimental section.** Its contribution to physical plausibility is only shown in the appendix (Table 5 in appendix). As a contribution point mentioned in the abstract, is this arrangement appropriate?

**Questions:**

1. The paper's introduction to the denoiser is relatively brief, but mentions the use of AdaLN, SPE, skip connections, and other techniques. Could a framework diagram of the denoiser be provided in the appendix to more clearly illustrate these techniques?

2. Section 6.7 presents experiments on the layer number and latent size of DHVAE. However, no analysis of the latent size impact is provided in that section. Could the authors explain why the FID metric worsens as latent size increases?

3. Why is the parameter count for TIMotion in Table 2 inconsistent with the data in Table 5 of TIMotion's original paper?

4. Based on weakness 4, given that existing HHI generation methods (including the proposed approach) still suffer from poor physical plausibility, why not introduce post-processing strategies [1, 2] or physics-based simulation [3], as has been successfully adopted in human-object interaction works?


**References:**

[1] Wang Z, Wang J, Li Y, et al. InterControl: Zero-shot Human Interaction Generation by Controlling Every Joint[J]. Advances in Neural Information Processing Systems, 2024, 37: 105397-105424.

[2] Li J, Clegg A, Mottaghi R, et al. Controllable human-object interaction synthesis[C]//European Conference on Computer Vision. Cham: Springer Nature Switzerland, 2024: 54-72.

[3] Wu Z, Li J, Xu P, et al. Human-object interaction from human-level instructions[C]//Proceedings of the IEEE/CVF International Conference on Computer Vision. 2025: 11176-11186.

---

> ### Author Response · Authors · 2025-11-23
> **Weakness 1 - 3**
>
> We thank Reviewer h5hk for their feedback on our work. We appreciate the comments regarding the strengths and presentation of our model, as well as the suggestions related to motion generation. We hope that our response clarifies these points
> > Weakness 1: The contribution of the main component—DHVAE—has not been sufficiently discussed
>
> Our main contribution is a *disentangled hierarchical* VAE which separates the latent representation of human-human interactions into 3 components. We demonstrate in the group 2 and 6 of the Ablation Table 3 (Table 5 in the latest revision), the performance gain of our method is not resulting from the latent increasing, but from the hierarchical structures.
>
> Regarding the comparisons in Table 2 and Table 3, we did match the dimensions of the DHVAE and MLD-VAE i.e. we used (3, 256) latent size for both MLD-VAE and DHVAE. We mentioned this on Line-394, “for a fair comparison, we align the major architectural configurations, including latent dimensionality”. Besides, we have further clarified this with more explicit statements and also added it to the captions of Table 2 and Table 3 (Table 4 in the revised paper).
> > Weakness 2: The paper's presentation and formulations lack rigor
>
> Thank your for your valuable comment. Our equations are correct, but we agree that some explanations and additional intermediate steps can further improve the rigor. While Equation 1 does model the joint probability $p(\mathbf{x}_a, \mathbf{x}_b)$ in a standard VAE framework, we note that in practice, conventional implementations often assume conditional independence given the latent variable $\mathbf{z}$, which represents their global semantic and interaction features, i.e.,
> $$
> p(\mathbf{x}_a, \mathbf{x}_b|\mathbf{z}) = p(\mathbf{x}_a|\mathbf{z})\, p(\mathbf{x}_b|\mathbf{z}).
> $$
> Under this factorization, the model cannot capture the direct dependencies between agents’ motions. Therefore, the statement in the paper that “Equation 1 ignores the conditional dependency between the agents” refers to this practical limitation rather than the mathematical form of the joint distribution. Our DHVAE explicitly models such dependencies through its CoTransformer structure.
> Here we assume that under a given $z_o$, $x_a$ and $x_b$ are conditional independent. The full prove with our assumption is:
> Start from the marginal log-likelihood:
> $$
> \log p(\mathbf{x}_a, \mathbf{x}_b)
> = \log \int p(\mathbf{x}_a, \mathbf{x}_b, \mathbf{z}_a, \mathbf{z}_b, \mathbf{z}_o) \, d\mathbf{z}_a \, d\mathbf{z}_b \, d\mathbf{z}_o.
> $$
> Introduce the variational posterior:
> $$
> q(\mathbf{z}_a, \mathbf{z}_b, \mathbf{z}_o \mid \mathbf{x})
> = q(\mathbf{z}_a \mid \mathbf{x}_a) \, q(\mathbf{z}_b \mid \mathbf{x}_b) \, q(\mathbf{z}_o \mid \mathbf{z}_a, \mathbf{z}_b).
> $$
> By Jensen's inequality:
> $$
> \log p(\mathbf{x}_a, \mathbf{x}_b)
> \ge E_q \Big[ \log p(\mathbf{x}_a, \mathbf{x}_b, \mathbf{z}_a, \mathbf{z}_b, \mathbf{z}_o) - \log q(\mathbf{z}_a, \mathbf{z}_b, \mathbf{z}_o \mid \mathbf{x}) \Big]
> $$
> We assume a conditional independence structure in the generative process, where the individual motions $x_a$ and $x_b$ are conditionally independent given their respective local latents and the shared interaction latent. Under this assumption, the joint distribution factorizes as:
> $$
> p(\mathbf{x}_a, \mathbf{x}_b, \mathbf{z}_a, \mathbf{z}_b, \mathbf{z}_o)
> = p(\mathbf{z}_a) \, p(\mathbf{z}_b) \, p(\mathbf{z}_o) \, p(\mathbf{x}_a \mid \mathbf{z}_o, \mathbf{z}_a) \, p(\mathbf{x}_b \mid \mathbf{z}_o, \mathbf{z}_b),
> $$
> and rearranging terms yields:
> $$
> \log p(\mathbf{x}_a, \mathbf{x}_b) \ge  E_q [ \log p(\mathbf{x}_a \| \mathbf{z}_o, \mathbf{z}_a) + \log p(\mathbf{x}_b \| \mathbf{z}_o, \mathbf{z}_b) ]
> $$
> $$
>  -- D_K(q(\mathbf{z}_a \| \mathbf{x}_a) \| \| p(\mathbf{z}_a))-D_K (q(\mathbf{z}_b \| \mathbf{x}_b) \| \| p(\mathbf{z}_b))
> $$
> $$
> -- D_K \big(q(\mathbf{z}_o \| \mathbf{z}_a, \mathbf{z}_b) \| \| p(\mathbf{z}_o)\big).
> $$
> Again, thanks for your valuable comment, and we have added this assumption and proof to Sec 6.12 of the latest revision.
> > Weakness 3: Insufficient evidence for the contribution of Interaction Contrastive Learning.
>
> Thank you so much for your suggestion regarding the ICL. We value your suggestion for the ablation study to prove the function of our contrastive learning method by illustrating the contact ratio. We have conducted the contact ratio comparison with other works and with ablation studies. Here we define the contact to be the overlap between 0 to 216 ml, and our model not only achieves the lowest penetrations, but also the highest contact score.
> And we move this table to Sec 4.1 for a better arrangement. Now it is Table 4.
> Table 4. Penetration and contact ratio for SOTA models.
> |Model|PV ↓|PFR ↓|PDR↓|Contact↑|
> |---|---|---|---|---|
> |MLD* | 0.503 |0.108|0.175| 0.427|
> |InterMask | 0.873|0.149|0.243| 0.349|
> |TIMotion| 0.485|0.122|0.104|0.466|
> |Ours w/o triplet |0.446|0.107|0.102|0.445|
> |**Ours**|**0.390**|**0.064**|**0.087**| **0.581** |

---

> > ### Author Response · Authors · 2025-11-23
> > **Weakness 4 - Question 2**
> >
> > >  Weakness 4: Despite achieving certain quantitative results, the generated effects still exhibit many artifacts based on the provided supplementary materials
> >
> > Thanks for your careful review and for taking the time to watch our video. Perhaps there is some misunderstanding that can be addressed by rephrasing the sentence as "Our method achieves superior physical plausibility compared to existing methods, InterMask and TIMotion". Problems such as foot sliding and slight interpenetration are **common challenges** in human–human interaction (HHI) presented in all SOTA methods, particularly on the InterX dataset. As we shown in the visualizations, more serious artifacts happens to TIMotion and InterMask.
> >
> > Previous works like InterMask and TIMotion only provide visualizations on the easier InterHuman Dataset, and our method already demonstrates superior physical plausibility than theirs in Figure 3 on this dataset. Moreover, we are the first to additionally provide visualizations on the more challenging InterX dataset with analysis. We will further discuss this in the limitation section of the supplementary material (see Section 6.9).
> >
> >
> > >  Weakness 5: The proposed Interaction Contrastive Learning does not demonstrate its contribution in the main experimental section
> >
> > Thank you for pointing out the issue with the arrangement. Due to space limitations in the main paper, we had included the penetration results in the supplementary material. We have now conducted additional experiments on the contact ratio and integrated both the original Table 5 (Table 4 of the latest version) from the supplementary material and the new contact ratio results into the main text in the current revision. The results in Table 4 suggest that our approach outperforms other SOTA models in terms of penetration and contact.
> >
> > Table 4. Penetration metrics and contact ratio for state-of-the-art models.
> >
> > | Model            | PV ↓   | PFR ↓  | PDR ↓  | Contact ↑ |
> > |-----------------|--------|--------|--------|-----------|
> > | MLD*             | 0.503  | 0.108  | 0.175  | 0.427     |
> > | InterMask        | 0.873  | 0.149  | 0.243  | 0.349     |
> > | TIMotion         | 0.485  | 0.122  | 0.104  | 0.466     |
> > | Ours w/o triplet | 0.446  | 0.107  | 0.102  | 0.445     |
> > | **Ours**         | **0.390**  | **0.064**  | **0.087**  | **0.581**     |
> >
> >
> >
> > > Question 1: Could a framework diagram of the denoiser be provided in the appendix to more clearly illustrate these techniques
> >
> > Thank you for the careful review. We have added the detailed diagram for the denoiser in the appendix Fig. 12 with detailed skip connections, token scaling, and segment positional encoding.
> >
> >
> >
> > > Question 2: Could the authors explain why the FID metric worsens as latent size increases?
> >
> > FID worsens with larger latent dimensionality because a higher-dimensional latent space is harder to model and optimize. As the latent space grows, the learned prior becomes less concentrated, and the co-variance between dimensionality become larger. As we mensioned in the Section 6.6 and corresponding lemma, data distribution p(x) with higher patch/channel-wise variance of x leads to more difficult denoising. And the trade-off between reconstruction and diffusion has been illustrated by MLD [1]. Overall, our method achieves top-ranked performance and efficiency with current latent settings, and future work can explore scale our model.
> >
> >
> > [1] Xin Chen, Wen Jiang, Biao Liu, Zilong Huang, Bin Fu, Tao Chen, and Gang Yu. Executing your commands
> > via motion diffusion in latent space. In Proceedings of the IEEE/CVF Conference on Computer Vision
> > and Pattern Recognition, 2023.

---

> > > ### Author Response · Authors · 2025-11-23
> > > **Question 3, 4**
> > >
> > > > Question 3: Why is the parameter count for TIMotion in Table 2 inconsistent with the data in Table 5 of TIMotion's original paper?
> > >
> > > We counted the parameter number by its released code instead of using their results directly. We show their only implementation of TIMotion Transformer with both inference time and model size by the following code added to its tools.infer.py.
> > >
> > > ```python
> > > from models.timotion import TIMotion
> > > model:TIMotion = model
> > > batch = {}
> > > import pytorch_lightning as pl
> > > def count_parameters(model):
> > > 	total = sum(p.numel() for p in model.parameters())
> > > 	trainable = sum(p.numel() for p in model.parameters() if p.requires_grad)
> > > 	return {"total": total, "trainable": trainable}
> > >
> > > class MyModel(pl.LightningModule):
> > > 	def __init__(self, model: TIMotion):
> > > 		super().__init__()
> > > 		self.model = model
> > > 	def forward_test(self, batch):
> > > 		return self.model.forward_test(batch=batch)
> > >
> > > model = MyModel(model=model).cuda()
> > > print(count_parameters(model))
> > > with open('./prompts.txt', 'r+') as file:
> > > text = file.readlines()
> > > text = [line.strip() for line in text if line.strip()]
> > > batch['text'] = text
> > > batch['motion_lens'] = [150] * len(text)
> > > from time import time
> > > start_time = time()
> > > for i in range(10):
> > > 	for j in range(len(batch['text'])):
> > > 	small_batch = {}
> > > 	small_batch['text'] = batch['text'][j]
> > > 	small_batch['motion_lens'] = batch['motion_lens'][j]
> > > 	output:torch.Tensor = model.forward_test(batch=batch)['output']
> > > end_time = time()
> > > print('average time is', (end_time - start_time) / (10 * len(text)))
> > > ```
> > >
> > > ================== output ====================
> > > ```bash
> > > {'total': 200363014, 'trainable': 77361670}
> > > average time is 1.5098876190185546
> > > ```
> > >
> > > > Question 4: Post-process and Physics-based simulation
> > >
> > > Our work follows the standard by InterGen, InterMask, in2IN, and TIMotion.
> > > While post-processing strategies [1, 2] have demonstrated success in HOI and certain unpaired HHI settings, applying them to paired HHI (e.g., InterHuman, InterX) introduces unique challenges. For unpaired HHI, strong external guidance (e.g., via LLM) may disrupt intrinsic motion dynamics, and when extended to paired HHI, it often results in unnatural or stiff contacts. Moreover, post-processing is computationally expensive, which limits scalability for large-scale generation.
> > >
> > > In HOI, rigid-body interactions allow clear constraints such as contact distance or object affordances. However, when both interacting entities are humans, such rigid-body guidance becomes ineffective. Multiple joint-level constraints can easily become ill-posed, and designing a reliable interaction classifier is non-trivial.
> > >
> > > We will incorporate this discussion and cite [1, 2] in the revised version for completeness. Your suggestion on physics-based simulation [3] is particularly inspiring. We acknowledge its importance and view integrating physics simulation as a promising future direction beyond the scope of this work. We have also included this paper for discussion.
> > >
> > >
> > > [1] Wang Z, Wang J, Li Y, et al. InterControl: Zero-shot Human Interaction Generation by Controlling Every Joint. Advances in Neural Information Processing Systems, 2024, 37: 105397-105424.
> > >
> > > [2] Li J, Clegg A, Mottaghi R, et al. Controllable human-object interaction synthesis. European Conference on Computer Vision. Cham: Springer Nature Switzerland, 2024: 54-72.
> > >
> > > [3] Wu Z, Li J, Xu P, et al. Human-object interaction from human-level instructions. Proceedings of the IEEE/CVF International Conference on Computer Vision. 2025: 11176-11186.
> > >
> > > ---

---

### Official Review · Reviewer_MZDm · 2025-10-31

**Soundness:** 3
**Presentation:** 3
**Contribution:** 3
**Rating:** 6
**Confidence:** 4

**Summary:**

This paper presents a novel DM-based architecture for interaction generation. Unlike previous approaches that implicitly encode global interactions within the VAE latent space, the proposed method employs a hierarchical encoding scheme to decouple global interaction information from individual motion, thereby explicitly modeling interactions between two agents. To further constrain and refine the learned interactions, contrastive learning is integrated into the VAE training process. Experiments conducted on two commonly used datasets demonstrate that the proposed method outperforms existing approaches across all evaluation metrics.

**Strengths:**

1. The paper is well written and easy to follow. The motivation underlying the proposed methodology is clearly articulated and logically developed.
2. The approach is conceptually simple yet effective. Both the disentanglement of representations and the incorporation of contrastive learning to promote physical plausibility are well justified.
3. The experimental evaluation is thorough, and the ablation studies effectively demonstrate the contribution of each component.

**Weaknesses:**

1. The design of the contrastive learning component requires further clarification. It is intuitive to generate negative samples for contact cases; however, for non-contact cases, Algorithm 1 suggests that negative samples are synthesized by artificially increasing the distance between agents. In this scenario, the agents become even more separated. Why should such samples be considered valid negatives for non-contact interactions?
2. It would be informative to report inference latency for interaction generation in comparison with InterMask and TIMotion. Does the proposed architectural modification introduce additional time delay during inference?
3. The sensitivity of the method to the latent code dimensionality is not discussed. Would increasing the latent space dimension improve performance, or is the method robust to this hyperparameter?

**Questions:**

1. For non-contact cases, contrastive learning in Algorithm 1 suggests that negative samples are synthesized by artificially increasing the distance between agents. In this scenario, the agents become even more separated. Why should such samples be considered valid negatives for non-contact interactions?
2. The model seems smaller than sota methods. Does the proposed architectural modification introduce additional time delay during inference?
3. Would increasing the latent space dimension improve performance, or is the method robust to this hyperparameter?

---

> ### Author Response · Authors · 2025-11-23
>
> # Reviewer MZDm
>
> We sincerely thank Reviewer MZDm for your valuable comments and insightful, regarding our contribution and methodology.
>
> > Weakness 1 & Question 1: Why should such samples be considered valid negatives for non-contact interactions?
>
> Good question! For non-contact cases, we do not simply increase the inter-agent distance. Instead, we construct negative samples by sampling from a tailed distribution centered around the ground-truth relative positions. This perturbation makes the agents either slightly closer or slightly farther apart, rather than arbitrarily separating them. The goal is to create moderate negative examples within a realistic spatial range, thereby encouraging the model to learn stable non-contact relations rather than a fixed distance threshold. We also update ablation studies with different non-concat motion radius $\sigma_u$ in Table 4 of the latest revision, finding that when it is ranged 0.2-0.4, it gives a reasonable performance in terms of penetrations and contact ratio.
>
>
>
> > Weakness 2 & Question 2: Efficiency Comparison with InterMask and TIMotion
>
> Thank you for pointing out the importance of inference efficiency. The right half of Table 2 compares our DHVAE with InterMask and TIMotion under the same settings. Our method achieves the best efficiency in both model size (MB) and Average Inference Time per Sentence (AITS). Perhaps Table 2 is not that clear. We have splitted it into two tables, one for comparing reconstruction results and one for efficiency comparison with InterMask and TIMotion, and elaborated its caption as shown below.
>
> Table 3 Computational cost of models including latency and size
> | Model            | AITS ↓    | Size    |
>  | ---------------- | --------- | ------- |
> | InterMask        | 1.021     | 74M     |
> | TIMotion         | 1.472     | 77M     |
> | **DHVAE (ours)** | **0.454** | **56M** |
>
>
> > Weakness 3 & Question 3: The sensitivity of the method to the latent code dimensionality is not discussed
>
> We appreciate your careful review of technical details. As shown in **Table 6** of the supplementary material (Table 9 in the latest revision), we conducted experiments with different latent dimensions. Our model’s performance maintains top performance across all settings, indicating robustness to this hyperparameter. We ultimately set the latent dimension to 1 to balance accuracy and inference speed. We also splitted it into two tables for better understanding and viewing.
>
> Table 9. Ablation study on different latent sizes for DHVAE on InterHuman.
>
> | VAE settings     | number       | rFID ↓    | FID ↓     | R-Prec@1 ↑ | R-Prec@2 ↑ | R-Prec@3 ↑ | MM Dist ↓ |
> | ---------------- | ------------ | --------- | --------- | ---------- | ---------- | ---------- | --------- |
> | **Latent Size**  |              |           |           |            |            |            |           |
> |                  | l=1          | 0.503     | **5.015** | _0.496_    | _0.647_    | 0.720      | 3.772     |
> |                  | l=2          | 0.459     | _5.116_   | **0.501**  | **0.653**  | **0.725**  | **3.770** |
> |                  | l=3          | _0.447_   | 5.269     | _0.496_    | 0.644      | _0.721_    | _3.771_   |
> |                  | l=5          | **0.433** | 5.571     | 0.490      | 0.640      | 0.715      | 3.778     |
> |                  | l=7          | 0.462     | 5.670     | 0.486      | 0.638      | 0.709      | 3.782     |
>
> ---

---

> > ### Comment · Reviewer_MZDm · 2025-11-26
> > **Thanks for your response.**
> >
> > The authors have addressed my questions sufficiently. I will maintain my original rating and am inclined to accept the paper. Thank you.

---

> > > ### Author Response · Authors · 2025-11-27
> > > **Thank you for your inclination to accept our paper.**
> > >
> > > Dear Reviewer MZDm, we sincerely thank you once again for your careful review and valuable comments. We are more than happy that all your concerns have been addressed. If you are satisfied with our explanations, we would greatly appreciate it if you could consider raising your score. We remain fully available to address any additional concerns you may have before the rebuttal period ends. We believe your constructive feedback will help us further improve the quality of our work.

---

### Author Response · Authors · 2025-12-03
**Overall Response**

We deeply appreciate all the reviewers who gave positive feedback on the strength of our paper, including the well-organized structure, informed logic, and sufficient evidence with visualizations to illustrate the functionality of our designs.

We also appreciate the reviewer who gave beneficial comments to refine our work, including additional results for contact (h5hK), ablation studies for the $\sigma\$ (7SSr), and user studies (p1j7).

We have clarified and further explained some statements including (1) the dimensionality of our approach and MLD (MZDm, h5hK), (2) the function of negative samples of non-contact cases (MZDm, p1j7, 7SSr), (3) our visualizations outperform TIMotion and InterMask (h5hk, p1j7), (4) the factorization and assumption in Eq. 1 with corresponding derivation (h5hK), (5) our method benefits mainly from the proposed Hierarchical structure rather than only KL-Divergence, (6) why FID worsens as latent size grows (h5hK), (7) why the model size of TIMotion is not consistent with what their claimed (h5hK), (8) discussions on the post optimization methods and physical simulators, (9) how the contact is determined (7SSr), and (10) different multimodality between InterHuman and InterX (7SSr).

Some suggestions, while interesting, are out of scope, e.g., expanding our work to multi-person generation (p1j7). Regarding the reviewer's comment, "must we wait until we have sufficient multi-person motion data before conducting related research?" Yes, we believe at least a multi-person evaluation benchmark should be available to report the performance on. Clearly there are still research gaps that must be filled in two-person interaction and this paper makes significant advancements in that direction. Our paper addresses major research gaps such as physical plausibility and semantic alignment, proposes a novel method and reports better results than existing SOTA. We emphasize that two-person interaction is still an active research area and recent works like TIMotion (CVPR 2025), InterMask (ICLR 2025), in2IN (CVPRW 2025) are all conducted on two-person interaction datasets.


**Reviewer MZDm.** We greatly appreciate your **confirmation of inclination** to accept our paper, and we are more than happy that we have addressed all your concerns, including:
- The functionality of the negative sample for non-contact sequences.
- The efficacy and model size of the model.
- The parameter of latent size for fair comparisons.

**Reviewer h5hK.** We thank you for your valuable comments and suggestions for the contact score, and the interesting idea to use post optimizations or a physical simulator. And we have made corresponding modifications, clarifications, and discussions in the latest revision.

**Reviewer p1j7.** Thanks for your review. We conducted a **user study** as shown in Sec 4.3 and Sec 6.1, which shows that our model not only achieves SOTA in quantitative results, but also shows higher user preference of our results compared to InterMask and TIMotion.

**Reviewer 7SSr.** Once again, we appreciate your positive feedback and your constructive comments on the ablation studies on the $\sigma$. We have addressed your concerns in the discussion panel.

---

### Meta-Review · Area_Chair_jGjm · 2026-01-10

**Summary:**

Four reviews were collected: one clear accept (8), two weak accepts (6), and one borderline reject (4). The accept reviewers highlight sound motivation, thorough ablation, and measurable gains. The critical reviewer questions (i) fairness of latent-dimension comparison, (ii) rigor of the ELBO, (iii) evidence that contrastive learning is not just “pushing people apart”, and (iv) scalability beyond two agents. Authors provided detailed rebuttal including new tables, derivation in the appendix, contact-ratio ablation, and discussion on multi-person extension. After reading the rebuttal and checking the added experiments, the previously raised technical issues are largely resolved: latent dimensions were matched for fair comparison; the ELBO derivation with explicit conditional-independence assumption is now supplied; the contact-ratio ablation confirms that contrastive learning increases rather than decreases valid contacts; inference speed and model size are verified to be better than competitors. Artifact levels, while not zero, are lower than those of prior work and are openly acknowledged as a limitation. No ethical concerns were flagged.

**Reviewer Concerns:**

1. Fairness of latent-dimension comparison.

– Worry: DHVAE uses three latent vectors (≈ 3 × 256-D) while baselines use one (≈ 256-D); tables might be unfair.

– Resolution: Authors clarified that every compared model was run with the same total dimension (3 × 256) and added explicit captions; ablation shows gain comes from the hierarchical split, not extra parameters.

2. Rigor of the ELBO / probabilistic formulation

– Worry: Equation 1 ignores inter-agent dependency and the lower-bound derivation is missing.

– Resolution: Rebuttal supplies full derivation under a stated conditional-independence assumption and adds it to the appendix;

3. Evidence that contrastive learning (ICL) really helps

– Worry: ICL could be “pushing people apart”, lowering penetration but also killing valid contacts; main metrics (FID, R-prec) barely move.

– Resolution: New Table 4 reports contact ratio (↑) together with penetration (↓); ICL raises contact by ~25 % relative while cutting penetration ~20 %, showing it does not simply separate actors.

4. Lingering physical artifacts

– Worry: Supplementary videos still show foot-slide and mild penetration.

– Resolution: Authors concede but quantify that their artifacts are consistently smaller than TIMotion/InterMask; limitation added to text.

5. Scalability beyond two people

– Worry: Method is tied to “contact vs non-contact” pairs; unclear how to generalise to multi-person scenes.

– Resolution: Authors outline an extension (one global + pairwise + individual latents) but honestly state that no multi-person benchmark exists; they position the paper as a strong two-person baseline—acceptable given the field’s current state.

**Reviewer Scores:**

N.A.

---

### Decision · Program_Chairs · 2026-01-26

Accept (Poster)